# Fate mapping of peripherally-derived macrophages after traumatic brain injury in mice reveals a long-lasting population with a distinct transcriptomic signature

Maria Serena Paladini[1,7], Benjamin A. Yang[1,7], Kristof A. Torkenczy[1], Elma S. Frias[1], Xi Feng[1], Karen Krukowski[1], Rene Sit[1], Maurizio Morri[1], Wendy Lam[1], Valentina Pedoia[1], Stefka Tyanova[1], Marco Colonna [2], Amber L. Nolan[3] & Susanna Rosi [1,4,5,6] ✉

Traumatic brain injury (TBI) is an environmental risk factor for dementia and long-term neurological deficits, posing a significant public health challenge. TBI-induced neuroinflammation involves both brain-resident microglia and peripheral monocyte-derived macrophages (MDMs). Previous research has shown that MDMs contribute to the development of long-term memory deficits, yet their long-term behavior following brain infiltration remains unclear. To address this, our study uses two complementary fate-mapping mouse lines, CCR2-creERT2 and Ms4a3-cre, for precise and lasting tracking of MDMs in vivo. Here we show that MDMs persist in the brain for at least 8 months post-TBI in both male and female mice. MDMs retain phagocytic activity for at least 30 days post-TBI, remain transcriptionally distinct from microglia, and display a gene expression profile associated with aging and disease. Moreover, we identify a core transcriptomic signature of MDMs shared across various mouse models and brain perturbations, which is also enriched in the brain myeloid cells of male subjects with TBI and Alzheimer's disease patients. These findings enhance our understanding of MDMs' dynamics after TBI and inform future targeted myeloid-based therapies.

Traumatic brain injury (TBI) presents a global health challenge as a leading cause of long-term physical, cognitive, and emotional disabilities, with no available treatment[1–3]. TBI exhibits a complex and multifaceted pathophysiology, beginning with a primary mechanical insult that induces secondary injuries from tissue and cellular damage that can last from hours to years[4,5], and long-lasting neurological damage in patients[6–8]. Key players in TBI-related neuroinflammation are resident macrophages localized in the brain parenchyma (further referred to as microglia) and infiltrating immune cells from the peripheral immune system, including monocytes and monocyte-derived macrophages (MDMs). Blood-circulating monocytes originate from hematopoietic stem cell progenitors in the bone marrow and can be

[1]Altos Labs, Redwood City, CA, USA. [2]Department of Pathology and Immunology, Washington University in St. Louis, School of Medicine, St. Louis, MO, USA. [3]Department of Laboratory Medicine and Pathology, University of Washington, Seattle, WA, USA. [4]Department of Physical Therapy & Rehabilitation Science, University of California, San Francisco, San Francisco, CA, USA. [5]Department of Neurological Surgery, University of California San Francisco, San Francisco, CA, USA. [6]Weill Institute for Neuroscience, University of California San Francisco, San Francisco, CA, USA. [7]These authors contributed equally: Maria Serena Paladini, Benjamin A. Yang. ✉e-mail: srosi@altoslabs.com

divided into two major subsets in mice: "classical" Ly6C[high]-Ccr2[+] monocytes, which can infiltrate inflamed tissues, and "non-classical" Ly6C[low]-Ccr2[-] monocytes that patrol the blood[9–11]. Ly6C[high]-Ccr2[+] monocytes are thought to be recruited to inflammatory sites, extravasate into the tissue, and differentiate into MDMs. The influx of monocytes into the injured brain is driven mainly by chemokines, a family of chemoattractant cytokines, that are released by brain-resident cells, including astrocytes and microglia[12,13]. CC ligand-2 (Ccl2, formerly MCP-1) and its receptor, Ccr2, are a well-established chemotactic pair in the context of TBI. Increased levels of Ccl2 have been found in the cerebrospinal fluid[14,15] and the serum[16] of human patients at acute and chronic time points after TBI, respectively. Ccl2 increases were also observed in the serum and in the lesioned cortex[14,16,17] of TBI mice. Moreover, blocking Ccr2 either genetically by using Ccr2[-/-] mice[14,18–21] or pharmacologically by using Ccr2 antagonists[16,22,23] dampens monocyte infiltration and prevents TBI-induced cognitive deficits in mice. MDMs have been explored as potential targets and mediators in TBI therapies, such as nanoparticle delivery[24,25] and vagus nerve stimulation[26]. While these prior studies aimed to establish a causal role for MDMs in neuroinflammatory damage and cognitive deficits following TBI, the long-term fate of engrafted MDMs remains unknown. The longitudinal trajectories of MDMs and their contribution to tissue-resident macrophages remain understudied, in addition to their tissue-specific function during homeostasis and disease[27]. In organs such as the heart, pancreas, and gut, circulating monocytes can give rise to and maintain the resident tissue macrophage population at steady state, while in the brain, microglia originate from the yolk sac and continue to self-maintain with minimal input from infiltrating peripheral cells[27–29]. However, during disease, specific chemokine signaling promotes the recruitment of peripheral monocytes into the brain parenchyma and their differentiation into macrophages. Some studies have suggested that monocytes invading the brain do not contribute to the resident microglia pool[30–32], while other groups have described that infiltrating monocytes transition to microglia[33,34]. A key limitation of these studies is the lack of reliable tools for precise cell tracing post-engraftment, which leaves the true ontogeny and identity of infiltrating monocytes and macrophages unclear. Recently developed mouse models that trace monocyte lineages are useful tools to contrast the roles of MDMs with those of microglia in disease and homeostasis[35]. Two leading models for monocyte tracing are driven by Ccr2 and Ms4a3 promoters, which are specifically expressed by monocyte-committed bone marrow progenitors[36–38], and have been used to study the long-term trajectories of macrophages during brain development[39], stroke[40], radiation[41], and neuroinflammation[37,38].

Here, we used Ccr2-creER[T2] and Ms4a3-cre mice to precisely trace blood-recruited MDMs at 7 days, 30 days, and 8 months after TBI, a chronic time point that has not been thoroughly investigated in prior studies, providing important knowledge of the long-term dynamics of MDMs after brain injury. The Ccr2-creER[T2] inducible Cre system allows to label MDMs in a time-controlled manner, specifically targeting the first wave of MDMs that infiltrated the brain acutely after TBI. In contrast, the Ms4a3-cre constitutive Cre system labels all MDMs infiltrating the brain, which provides a more comprehensive view of MDMs dynamics. Together, these systems enabled us to assess the spatial localization, transcriptomic signatures, and phagocytosis activity of MDMs with respect to microglia at sub-acute and chronic time points. Moreover, we identified a core transcriptomic signature of infiltrated macrophages across mouse experimental models and validated it in a single-nucleus RNA sequencing (sn-RNAseq) dataset generated from human orbitofrontal cortex tissues of TBI patients, as well as in the Alzheimer's Disease sn-RNAseq dataset from the Seattle Alzheimer's Disease Brain Cell Atlas (SEA-AD)[42]. These findings provide important knowledge on MDMs and will be valuable for developing therapeutic strategies aimed at specifically targeting these cells.

## Results

### MDMs infiltrate the injured brain acutely after TBI and persist for at least 8 months

To investigate the fate of MDMs after TBI, Ccr2-creER[T2] mice were crossed with Ai14D mice to enable specific, temporally-controlled labeling of Ly6C[high]-Ccr2[+] monocytes upon tamoxifen administration[37,38]. Given the fast turnover of Ly6C[high] monocytes[43], we first tested the labeling efficiency of Ly6C[high] monocytes in the blood after 5 daily tamoxifen injections by flow cytometry (Fig. 1a). Starting from 3 doses of tamoxifen and up to two days after stopping treatment (2 days off), the labeling efficiency of tdTomato+ Ly6C[high] monocytes exceeded 80% (Fig. 1b). Of note, due to the rapid turnover of Ly6C[high] monocytes, the fraction of labeled Ly6C[hi] monocytes decreased to around 25% one week after the last tamoxifen injection, and to almost zero after two weeks (Fig. 1b). These results demonstrate that the Ccr2-creERT2::Ai14D system enables precise, temporally-controlled labeling of specific monocyte populations.

We have previously shown that the accumulation of MDMs in the ipsilateral hemisphere starts at 12 hours and peaks at 24 hours after TBI[16]. To confidently capture and follow the initial infiltration window of Ly6C[high] monocytes, the Controlled Cortical Impact (CCI) model was used to induce TBI in mice that received three out of the five total doses of tamoxifen (Fig. 1c). At this time point (0dpi, 3 tamoxifen doses), the labeling efficiency was between 80 – 85%, with no sex differences observed (Supplementary Fig. 1a). Of note, off-target labeling of neutrophils (Ly6C + Ly6G+) and patrolling monocytes (Ly6C[low]) was negligible (Supplementary Fig. 1b). We next evaluated TBI-induced infiltration and engraftment of labeled Ccr2+ cells by flow cytometry of whole brain samples at different time points (Fig. 1d). As expected, we measured labeled MDMs in the TBI brain after 7 days (average count: 5890.5 ± 490 cells, 8.2 ± 0.81% of the CD11b + CD45+ whole brain population), in line with our previous work[16]. Surprisingly, MDMs that infiltrated acutely after TBI were still present after 1 month (average count: 1862 ± 87 cells, 1.52 ± 0.08 % of the CD11b + CD45+ whole brain population) and 8 months (average count: 580.3 ± 92 cells, 0.71 ± 0.07% of the CD11b + CD45+ whole brain population) (Fig. 1e, f) post-TBI. Importantly, no significant off-target labeling of microglia (CD11b+CD45[mid/low]) was detected in TBI mice (Supplementary Fig. 1c), and no labeling of CD11b + CD45+ cells was detected in sham mice (Supplementary Fig. 1d). These data provide the first evidence that engrafted MDMs can last in the brain for several months following TBI.

### TBI-induced cognitive deficits last for at least 8 months in Ccr2-creER[T2]::Ai14D mice

Given the long-lasting presence of infiltrated MDMs and previous work demonstrating their causal role in the development of TBI induced cognitive deficits[16,19], we measured learning and memory function at 7 days, 30 days and 8 months after TBI using the radial arm water maze (Fig. 1g). As previously reported[44,45], mice were first trained to locate a platform hidden under opaque water placed in one of the eight arms using navigational cues placed in the room over one or two learning days of 9 or 6 trials, respectively (see Methods). Spatial memory was then tested by running a memory probe (3 consecutive trials) twenty-four hours and one week after training. The number of non-target arm entries before locating the escape platform (errors) was used as a measure of learning and memory deficits. Errors from three consequent trials were averaged in blocks. Of note, no sex differences were observed in the RAWM performance of female and male mice (Supplementary Fig. 2) and were therefore plotted together. In line with previous studies, TBI induced a significant impairment in learning and memory as early as one week after TBI (Two-way RM ANOVA, TBI effect: $F_{1,30} = 9.783$, $p = 0.0039$; time effect: $F_{2.654,71.66} = 23.33$, $p < 0.0001$) compared to sham mice (Fig. 1h). Cognitive performance continued to

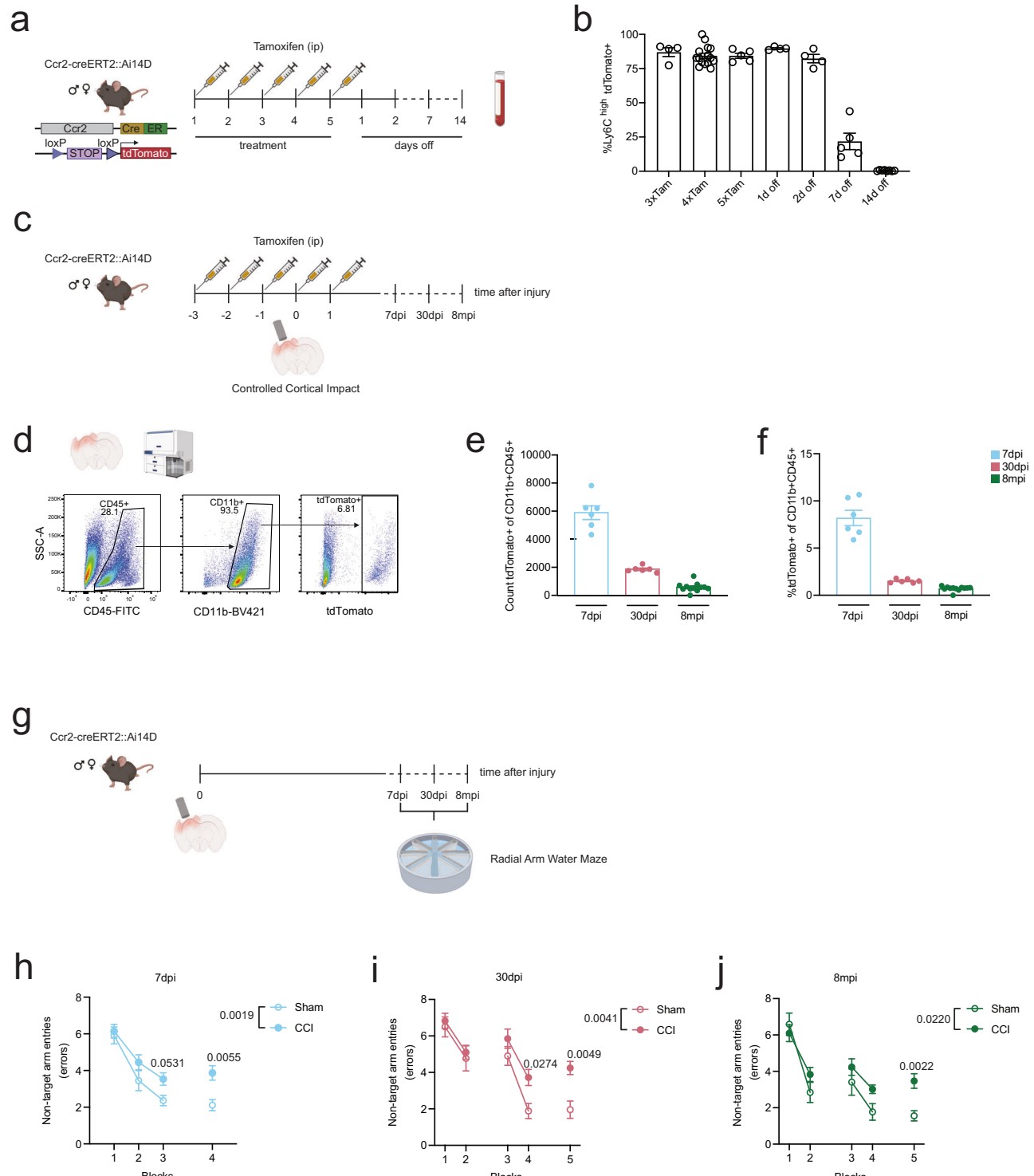

be affected in mice at one month after TBI (Two-way RM ANOVA, TBI effect: $F_{1,28}$ = 9.803, $p$ = 0.0041; time effect: $F_{3,120,87.35}$ = 23.41, $p < 0.0001$) (Fig. 1i). Remarkably, cognitive deficits persisted in TBI Ccr2-creER$^{T2}$::Ai14D mice for up to 8 months (Two-way RM ANOVA, TBI effect: $F_{1,32}$ = 5.792, $p$ = 0.0220; time effect: $F_{3.385,108.3}$ = 27.18, $p < 0.0001$) (Fig. 1j). These results demonstrate that Ccr2-creER$^{T2}$::Ai14D female and male mice develop early and long-lasting deficits in spatial learning and memory after TBI. Notably, we demonstrate that TBI-induced cognitive deficits as measured by RAWM persist for up to 8 months in injured mice.

## MDMs accumulate in the pericontusional region and undergo phenotypic transition

To map the spatial localization of MDMs in the TBI brain, we acquired z-stacked tiled images of the pericontusional area at 3 different coordinates from rostral to caudal[46,47] (Fig. 2a). As shown in the representative images (Fig. 2b), tdTomato+ MDMs were concentrated in the cavitation area, with a small fraction lining the third ventricle, choroid plexus, and CA1/DG regions of the hippocampus at 7 days after TBI. At chronic time points, cells were mainly localized in the thalamus under the ipsilateral hippocampus and in the choroid plexus (Fig. 2b). In line

**Fig. 1 | MDMs accumulate and persist in the mouse brain for at least 8 months after TBI, paralleled by long-lasting cognitive deficits. a** Experimental design - tamoxifen-induced labeling of Ly6C$^{hi}$-Ccr2+ blood monocyte in Ccr2-creER$^{T2}$::Ai14D mice. Blood from the tail vein was collected at different time point and analyzed by flow cytometry. **b** Percentage of blood Ly6C$^{hi}$ monocytes efficiently labeled (tdTomato + ) in male and female Ccr2-creER$^{T2}$::Ai14D mice. Individual animals are plotted (3xTam $n$ = 4, 4xTam $n$ = 15, 5xTam $n$ = 5, 1 d off $n$ = 4, 2 d off $n$ = 4, 7 d off $n$ = 5, 14 d off $n$ = 8). Data are expressed as the mean of the examined variable ± SEM. **c** Experimental design - female and male Ccr2-creER$^{T2}$::Ai14D mice received 5 doses of tamoxifen and were injured using the Controlled Cortical Impact model of TBI on the fourth day of tamoxifen treatment. **d** Gating strategy for quantifying Ccr2+ (tdTomato+) cells from the TBI brain of Ccr2-creER$^{T2}$::Ai14D mice at 7 and 30 days post injury (dpi) and at 8 months post TBI injury (mpi). **e** Average cell number of Ccr2+ (tdTomato + ) cells in the CD11b + CD45+ population **f** Percentage of Ccr2+ (tdTomato + ) cells in the CD11b + CD45+ population. Individual animals are plotted

(7dpi $n$ = 6, 30dpi $n$ = 6, 8mpi $n$ = 12). Data are expressed as the mean of the examined variable ± SEM. **g** Experimental design - Ccr2-creER$^{T2}$::Ai14D mice were tested in the Radial Arm Water Maze (RAWM) task to detect TBI-induced cognitive deficits at 7 and 30dpi and at 8mpi. **h–j** Injured Ccr2-creER$^{T2}$::Ai14D mice made significantly more errors when performing the RAWM task compared to sham (Two-way RM ANOVA revealed TBI effect $p$ = 0.0019 and time effect $p$ < 0.0001 at 7dpi; TBI effect $p$ = 0.0041 and time effect $p$ < 0.0001 at 30dpi and TBI effect $p$ = 0.0220 and time effect $p$ < 0.0001 at 8mpi). 7dpi $n$ = 12 males, 6 females; 30dpi $n$ = 7 males, 12 females; 8mpi $n$ = 11 males, 13 females. Data are expressed as the mean of the examined variable ± SEM. Statistical differences from multiple comparison tests are denoted in the graphs. (Two-way RM ANOVA with Šidák multiple comparisons test, shown in the figure). Source data are provided as a Source Data file. Created in BioRender. Krukowski, K. (2025) https://BioRender.com/36oc46m.

with the cell numbers observed by flow cytometry, we observed fewer tdTomato+ infiltrated macrophages at 30dpi and 8mpi (Fig. 2b). To determine if infiltrated tdTomato+ cells were differentiating into microglia, we analyzed the colocalization of tdTomato+ cells with ionized calcium-binding adapter molecule1 (Iba1) (a pan-myeloid cells marker) and the purinergic receptor P2ry12 (a microglia specific marker) across the top quarter of the mouse brain coronal section (peri-contusional region) using Imaris, a microscopy image analysis software (Fig. 2c). We found a significant increase in the fraction of tdTomato+ cells that expressed Iba1 over time after engraftment (One-way ANOVA, $F_{2,11}$ = 51.17, $p$ < 0.0001, Fig. 2d, Supplementary Fig. 3). Notably, no tdTomato+ Iba1+ cells expressed P2ry12 at 7dpi, but we observed a trend towards increased P2ry12 expression, though not statistically significant, at 8mpi compared to 30dpi (Fig. 2e, Supplementary Fig. 3). Next, we analyzed morphological changes in cell volume across the 3 selected hippocampal regions (anterior, central and posterior, Fig. 2a). We observed a general reduction in cell volume from 7dpi to 30dpi and 8mpi in tdTomato+ cells that also express Iba1+ (One-way ANOVA, Anterior $F_{2,1706}$ = 270.7, $p$ < 0.0001, Central $F_{2,2147}$ = 184.9, $p$ < 0.0001, Posterior $F_{2,1693}$ = 322.6, $p$ < 0.0001) (Fig. 2f). Interestingly, tdTomato+Iba1+P2ry12+ cells' volume increased from 30 days to 8 months after engraftment (Fig. 2g). No tdTomato+ cells were found in the brains of sham mice or in the contralateral hemispheres of TBI mice at any time points (Supplementary Fig. 5a, b). These results show that MDMs preferentially accumulate in the TBI region, undergo morphological changes, and acquire the pan-myeloid cells marker Iba1 over time after engraftment. However, they do not fully differentiate into microglia.

## MDMs maintain their phagocyte competency after engrafting the TBI brain

Phagocytosis is an essential function of macrophages. To test the phagocytic capacity of TBI-induced MDMs and examine how this function changes over time after infiltration, we used our well-established in vivo phagocytosis assay[44,48]. Blue fluorescent beads (2μm diameter) were injected into the ipsilateral hippocampi of TBI Ccr2-creER$^{T2}$::Ai14D mice on days 4 and 27 after TBI, and the brains were harvested three days later (7dpi or 30dpi) for IF analysis (Fig. 2h). Three-dimensional (3D) reconstruction of MDMs engulfing beads is shown in Fig. 2i. TdTomato+ MDMs and blue beads were automatically segmented in maximum intensity projections of z-stacks from 40x tiled images of the pericontusional region (Fig. 2i, j). We then quantified the percentage of tdTomato+ MDMs that engulfed one or more fluorescent beads. Strikingly, macrophages' capability to phagocytose synthetic beads was unchanged between 7 and 30dpi (Fig. 2k). Of note, no sex differences were observed (Supplementary Fig. 5). These data demonstrate that MDMs that engraft the brain after TBI can phagocytose both at subacute and chronic time points.

## Transcriptional profiles of MDMs after engraftment are persistent, distinct from microglia, and consistent with signatures of aging, senescence, and disease

After establishing the persistence of MDMs in the TBI brain, we investigated how the transcriptomes of these cells change over time after TBI. MDMs (tdTomato + ) were sorted from the CD11b + CD45+ cell population in the whole Ccr2-creER$^{T2}$::Ai14D brain at 7 days, 30 days, and 8 months after TBI for bulk RNA sequencing. Microglia were sorted as CD11b + , CD45$^{mid/low}$, tdTomato- at 8mpi (Fig. 3a, Supplementary Fig. 6). To characterize the molecular phenotypes of each group, we scored each sample by its enrichment of Reactome pathways[49] using the singscore package[50] and performed sparse partial least squares linear-discriminant analysis (sPLS-DA) of the scores to identify pathways that distinguish each group. The first discriminant component captured the temporal dynamics of MDMs after brain engraftment, while the second component highlighted a distinct separation between MDMs and microglia at 8 months post-TBI, and the third component separated MDMs at 7 days and 30 days after TBI (Fig. 3b, Supplementary Fig. 7). These results suggest that MDMs collected at 8 months after infiltration exhibit an intermediate transcriptional profile between that of microglia and MDMs at earlier time points. At 7 days post-TBI, MDMs were enriched for processes related to immune signaling and cell stress responses that transitioned to responses to neuronal signaling and maresin signaling at 30 days, and MDMs at both 7 and 30dpi upregulated pathways related to histone modifications and DNA damage relative to the 8mpi (Fig. 3c). At 8 months, MDMs upregulated terms related to immune system activation (e.g., Complement cascade, Interleukin-2 signaling, Synthesis of Leukotrienes (LT) and Eoxins(EX)), cell-cell interactions (e.g., Regulation of CDH11 Expression and Function, Regulation of Homotypic Cell-Cell Adhesion, Signaling to RAS), and signal transduction (e.g., Cell death signaling via NRAGE, NRIF and NADE) relative to resident microglia, which were instead enriched for terms related to cell maintenance (Receptor Mediated Autophagy and Cilium Assembly) (Fig. 3c). We observed similar separation between groups when we performed the same analysis with the ImmunoSigDB[51] database, a compendium of 5000 gene-sets from multiple immunology studies (Supplementary Fig. 7). Differential gene expression testing between MDMs and microglia at 8 months after TBI showed that MDMs expressed higher levels of *Ccl2, Ccr2, Lyz2* and F4/80 (*Adgre1*) and lower levels of microglia marker genes such as *Sall1, Hexb, Tmem119* and *P2ry12* (Fig. 3d, e). These data demonstrate that MDMs retain distinct transcriptomic profiles even long after brain engraftment.

Next, we compared the MDM transcriptomes to recent, well-established signatures of macrophages and microglia. We focused on the "Disease Inflammatory Macrophages" (DIM) population, which accumulates in the mouse brain with aging and in neurodegenerative conditions[41], and on the "Disease Associated Microglia" (DAM) signature[41,52]. As shown in Fig. 3f, both DIM and DAM signatures were

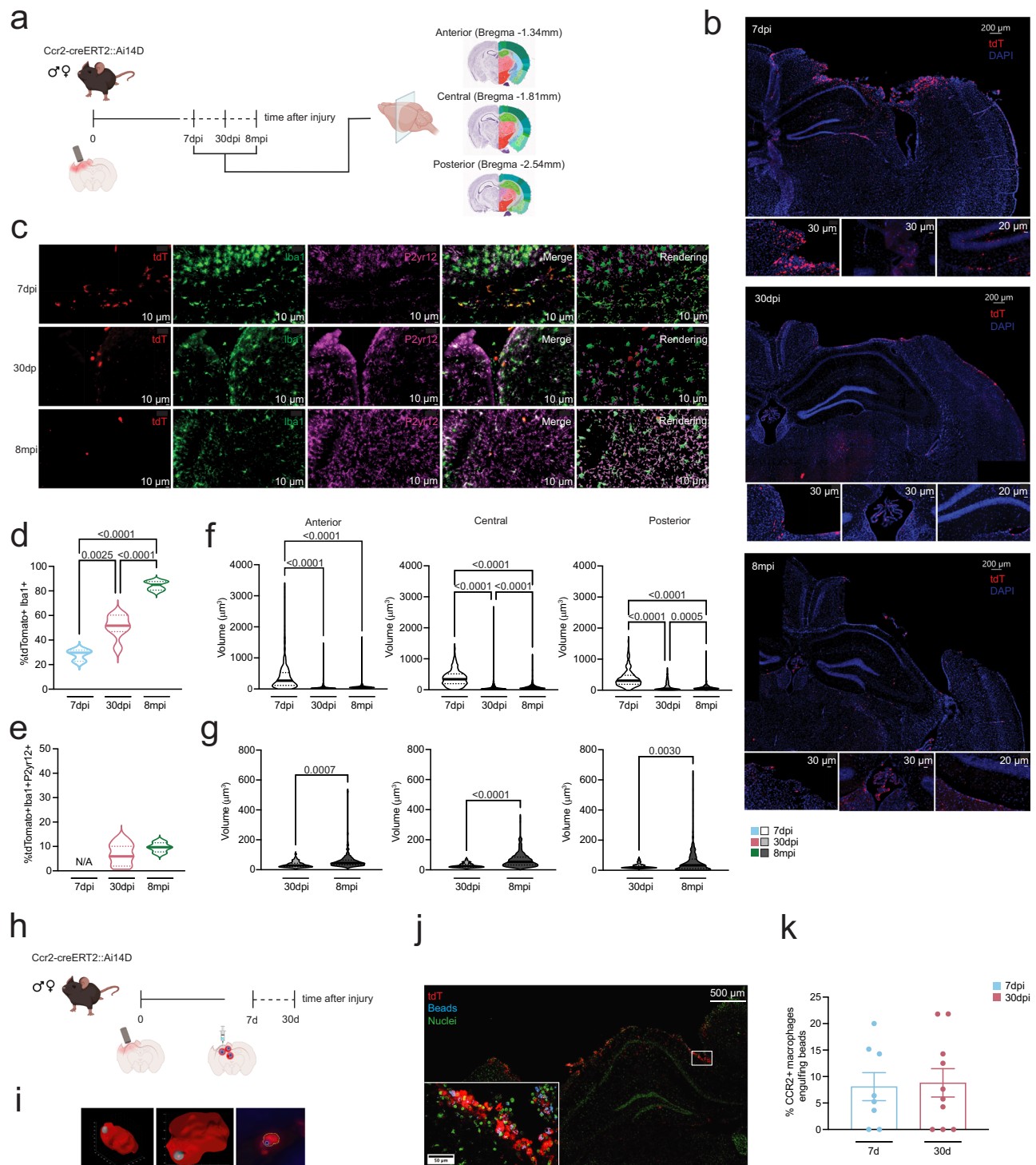

enriched in MDMs, especially at early time points after TBI, in contrast to microglia (Fig. 3f). Next, we compared our data with the SenMayo gene set, a list of senescence-associated genes that was validated in aged human and mouse samples[53], and found that MDMs upregulated genes related to senescence relative to microglia (Fig. 3f). Similar results were observed when comparing our infiltrated macrophages with microglia isolated from 24 months old mice ("Microglia Aging") and from Ercc1Δ/KO mice that displayed features of accelerated aging ("Accel. Microglia Aging")[54] (Fig. 3f). Finally, we assessed the gene expression of *Cdkn1a (p21)* and *Cdkn2a (p16)*, two of the most widely used genetic markers of senescence that were not included in the SenMayo signature[53,55]. As shown in Fig. 3g, mRNA levels of *p21* and *p16*

were significantly higher in MDMs at 8 months after TBI when compared to microglia. These results reveal that the transcriptomic signature of MDMs is associated with aging, senescence, and disease, distinct from microglia.

## MDMs share a common core transcriptomic signature across fate-mapping mouse models and brain perturbations, enriched in human brain myeloid cells from TBI and AD patients

We validated our transcriptomic findings using Ms4a3-cre::Ai14D mice, a fate mapping model complementary to the Ccr2-creER[T2] system[36]. In this model, all progeny of granulocyte monocyte progenitors (GMP), including Ly6C[hi] monocytes, are constitutively labeled[37] (Fig. 4a). Of

**Fig. 2 | MDMs infiltrate the pericontusional region and maintain their phago-cytic ability after engraftment. a** Experimental design - Ccr2-creER[T2]::Ai14D mice were injured using the controlled cortical impact TBI model and brain samples were collected at 7 and 30 days post injury (dpi) and at 8 months post TBI injury (mpi). Pericontusional regions (top quarter of a coronal brain section) were acquired as 20x z-stacked tiled images. Cavitation images were acquired at 3 different coordinates from Bregma: −1.34 mm (Anterior), −1.81 mm (Central) and −2.54 mm (Posterior). Nissl (left) and anatomical annotations (right) from the Allen Mouse Brain Atlas and Allen Reference Atlas – Mouse Brain. Allen Mouse Brain Atlas, mouse.brain-map.org and atlas.brain-map.org. Anatomical annotations from the Allen Reference Atlas – Mouse Brain, https://atlas.brain-map.org/. **b** Representative images of pericontusional regions of Ccr2-creER[T2]::Ai14D mice at different time points. **c** Representative example of machine learning-based surface rendering (Imaris 10.2.0) at different time points. **d** % of tdTomato+ cells that express Iba1. **e** % of tdTomato+ cells that express Iba1 and P2ry12. **f** Volume (μm³) of tdTomato+Iba1+ cells across the 3 selected coronal sections. **g** Volume (μm³) of tdTomato +Iba1+P2ry12+ cells across the 3 selected coronal sections. Violin plots depict the distribution of samples. 7dpi $n = 2$ males, 1 female; 30dpi $n = 3$ males, 4 females; 8mpi $n = 2$ males, 2 females. (One-way ANOVA and two-sided unpaired Student's $t$-test, statistical difference are denoted in the graphs). **h** Experimental design - Ccr2-creER[T2]::Ai14D mice were injured using the controlled cortical impact TBI model and phagocytosis capacity was measured at 7 and 30dpi. Fluorescent beads (2μm diameter) were injected in the ipsilateral hippocampus, and brains were collected 3 days after for IF analysis. **i** 3D rendering of MDM engulfing one bead and K-means segmentation overlay. **j** Representative confocal image of MDMs (tdTomato, red) engulfing injected beads (blue) in the pericontusional area. Nuclei are visualized in green. **k** % of tdTomato+ macrophages that engulfed one or more fluorescent beads was quantified for each acquired image. Each dot is the average of 2/4 images acquired for each animal. Individual animals are plotted (7dpi $n = 4$ males, 4 females; 30dpi $n = 5$ males, 5 females). Data are expressed as the mean of the examined variable ± SEM. (Two-sided unpaired Student's $t$-test). Source data are provided as a Source Data file. Created in BioRender. Krukowski, K. (2025) https://BioRender.com/uoqnpdv.

note, Ms4a3-cre::Ai14D mice displayed TBI-induced cognitive impairments in RAWM similar to Ccr2-creER[T2]::Ai14D mice (Supplementary Fig. 8). MDMs (tdTomato + ), microglia (CD11b+, CD45[mid/low], tdTomato-) and "classical" monocytes (Ly6G-, Ly6C[hi]) were sorted from the brain and blood of female and male Ms4a3-cre::Ai14D mice at 7 and 30 days post TBI (Fig. 4a, Supplementary Fig. 6). Ms4a3+ MDMs displayed higher levels of *Ccr2* mRNA compared to microglia, comparable with what we observed in the Ccr2-creER[T2] dataset (Fig. 4b). Microglia-specific genes were enriched in microglia versus Ms4a3+ MDMs at both time points, confirming that infiltrated macrophages retain their identity (Fig. 4c–f). We corroborated the enrichment of transcriptomic signatures associated with aging, senescence, and disease in Ms4a3+ MDMs in comparison to sham microglia, with the exception of the DAM signature (Fig. 4g). We then combined the transcriptomic data of microglia, monocytes, and MDMs from the Ccr2 and Ms4a3 datasets, and mapped them onto cell type signatures from the Molecular Signatures Database (MSigDB)[56] using singscore. sPLS-DA analysis of the resulting scores revealed general clustering by cell identity and separation by time point (Fig. 4h, Supplementary Fig. 9). Strikingly, these results confirmed that MDMs from two different genetic models are transcriptomically distinct from microglia and monocytes, regardless of how long they have been engrafted in the brain. Next, we investigated if MDMs share a core genetic signature across different models and experimental conditions. For this purpose, we compared our datasets with other bulk transcriptomic datasets of brain-engrafted macrophages available in the literature[30,48]. We identified 114 genes that were commonly upregulated in MDMs (Supplementary Data 1) (Fig. 4i). These genes were enriched for Gene Ontology Biological Processes related to immune response pathways such as antigen processing and presentation, immune cells activation, and differentiation (Fig. 4j). Of note, no overlap was observed between the MDMs core signature and DIM/DAM signatures (Supplementary Fig. 10). These results strengthen the concept of long-lasting MDMs as distinct cell populations that share a core signature that is consistent across different models and experimental conditions.

To explore how our observations relate to the human brain, we performed single-nuclei RNA sequencing (snRNA-seq) on orbitofrontal cortex tissue obtained post-mortem from male patients who had sustained TBI several years prior to their death, as well as age- and sex- matched non-TBI controls. The tissue samples were obtained from the Pacific Northwest Brain Donor Network (University of Washington). Detailed demographic and clinical characteristics are provided in Supplementary Table S1. It is worth noting that no standardized annotation for infiltrating macrophages has yet been established in the human brain. We used Azimuth[57] to align our data to the human cortex reference and annotate cells. We then assessed the enrichment of the human orthologous genes of our MDMs core signature within the human dataset (Supplementary Data 1). Remarkably, we observed significant enrichment of MDMs signature in the human Micro-PVM populations of TBI patients, which include both microglia and macrophages (Figs. 4k, l Supplementary Fig. 11). This finding provides evidence that the MDM signature we derived from mouse TBI tissue is conserved in the human brain's myeloid cell population and is significantly upregulated in TBI patients compared to controls. Lastly, we questioned whether the MDMs signature is conserved across different diseases featured by cognitive deficits. To address this, we utilized the Seattle Alzheimer's Disease Brain Cell Atlas (SEA-AD)[42], a comprehensive resource that provides molecular data on Alzheimer's disease (AD), including gene expression profiles from brain tissue samples of both AD patients and healthy controls. We observed a strong enrichment of our signature in the Micro-PVM cell populations of AD patients with dementia when compared to donors with normal cognitive function (Fig. 4m, n). Although we also observed an enrichment in astrocytes and OPCs from AD patients, it was not as pronounced as in the Micro-PVM populations (Fig. 4m, n). This shows that the MDMs signature is conserved in myeloid cells across species and across diseases, and suggests that it may be a common feature in neuroinflammatory processes associated with cognitive decline.

## Discussion

Using two independent and complementary fate mapping systems, we described a population of MDMs that persist in the mouse brain parenchyma for at least 8 months after TBI. We then identified a MDM's core transcriptomic signature shared across different mouse models and brain perturbations. Importantly, this signature was also found in human brain samples of individuals years after TBI, as well as in AD patients.

While we have previously demonstrated the causal role of MDMs in the development of memory deficits following TBI[16,19], the fate of MDMs after engraftment remains unknown. We and others have previously shown that the accumulation of these cells in the TBI brain is temporally restricted[58,59], as these cells virtually disappear from the brain parenchyma after a few days[16]. The explanation behind this phenomenon lies in the use of Ccr2[RFP/+] knock-in mice, where infiltrated macrophages will stop expressing Ccr2 and therefore their fluorescent tag, after in situ reprogramming[40]. Lineage tracing using fate mapping models is a useful tool to unravel the long-term trajectories of MDMs[35]. By using this approach, we were able to show the presence of labeled MDMs in the mouse brain for at least 8 months post-TBI, challenging the assumption about their transient nature[58]. Of note, the number of infiltrated macrophages that we detected subacutely (7dpi) in injured mice (around 6000/whole brain) was higher than what we published before in CX3CR1[+/GFP]Ccr2[+/RFP] mice (less than 500/ipsilateral

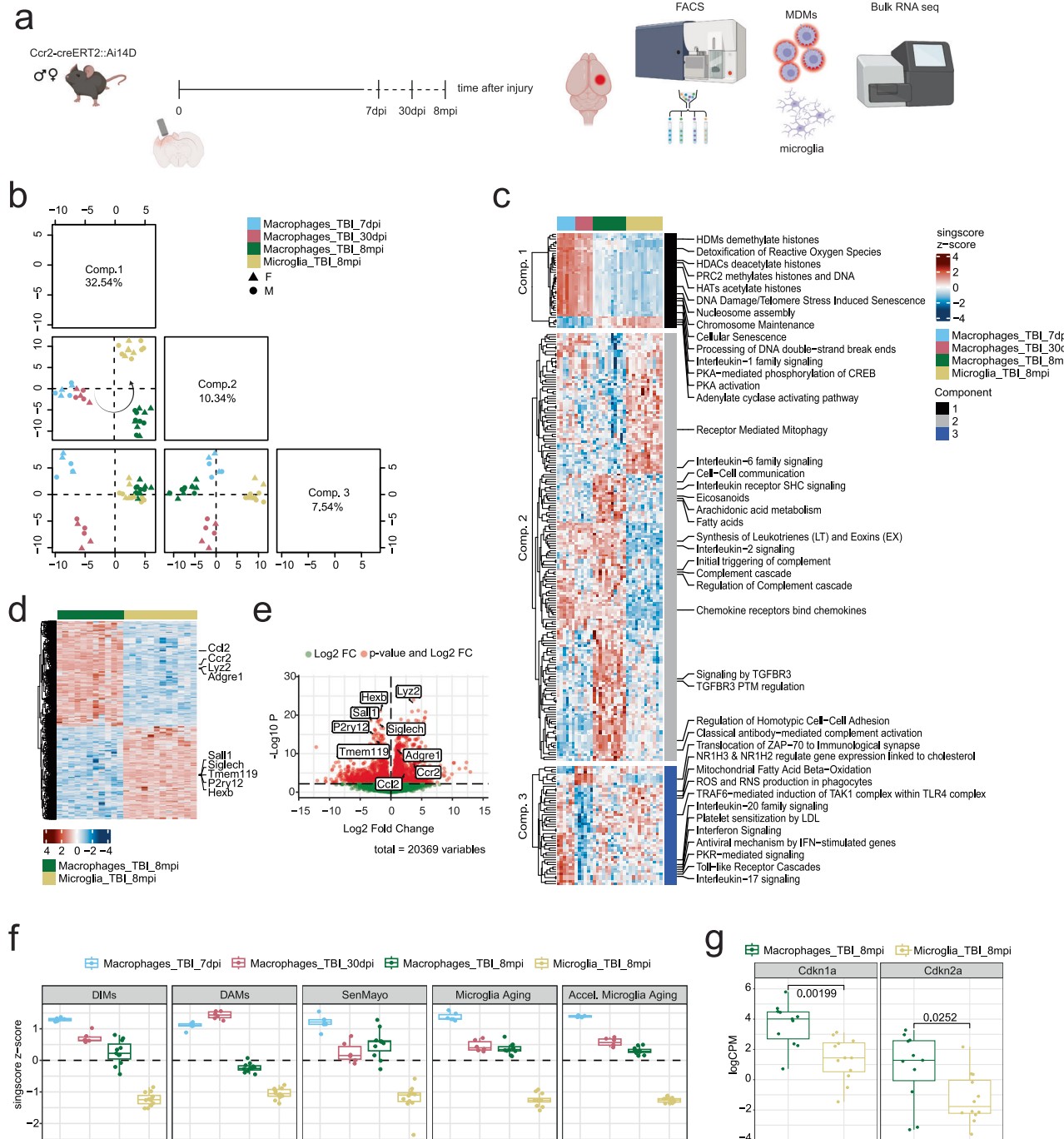

**Fig. 3 | Transcriptomic signatures of MDMs after TBI. a** Experimental design - Ccr2-creER[T2]::Ai14D mice were injured using the controlled cortical impact traumatic brain injury (TBI) model and brain samples were collected at 7 and 30 days post injury (dpi) and at 8 months post TBI injury (mpi). Microglia (tdTomato-) and monocyte-derived macrophages (MDMs) (tdTomato+) cells were isolated from the CD11b + CD45+ brain cell population by fluorescence-activated cell sorting (FACS) and processed for bulk RNA sequencing. **b** Projections of samples onto the first three components from sparse partial least squares discriminant analysis (sPLS-DA) of Reactome singscore scores from TBI MDMs at 7dpi, 30dpi, and 8mpi, with 8mpi TBI microglia. **c** Heatmap of Reactome singscore z-scores for significant terms for each component selected by sPLS-DA. The top terms for discriminating each group are shown. **d** Heatmap and **e** volcano plot of differentially expressed genes from TBI MDMs and microglia at 8mpi. **f** Reactome singscore z-scores for reference signatures of disease-inflammatory macrophages (DIMs), disease-associated microglia (DAMs), SenMayo, Microglia Aging, and Accelerated Microglia Aging (see methods). **g** Expression (logCPM) of *Cdkn1a (p21)* and *Cdkn2a (p16)* in TBI MDMs and microglia 8 months after injury (two-sided Welch's t-tests with Holm multiple testing correction, *p-value* in the graph, *Cdkn1a* df=20.83321, *Cdkn2a* df=17.89013). Individual animals are plotted (MDMs: 7dpi *n* = 3 females, 3 males, 30dpi *n* = 3 females, 3 males; 8mpi *n* = 5 females, 6 males. Microglia 8mpi: *n* = 6 females, 6 males). Box plots depict data quartiles. Source data are provided as a Source Data file. Created in BioRender. Krukowski, K. (2025) https://BioRender.com/7s7n0z7.

hemisphere at peak accumulation)[16], and what Doran et al., found one week after TBI in WT mice (around 1000 cells/ipsilateral hemisphere)[60]. This disparity can be explained by the different mouse models deployed, and demonstrates the significant advantages of using a more rigorous fate mapping system. The number of macrophages that persist in the brain after infiltration decreases from subacute time points (7dpi, around 6000 cells) to chronic time points (30dpi, around 2000 cells; 8 mpi, around 600 cells). Leveraging

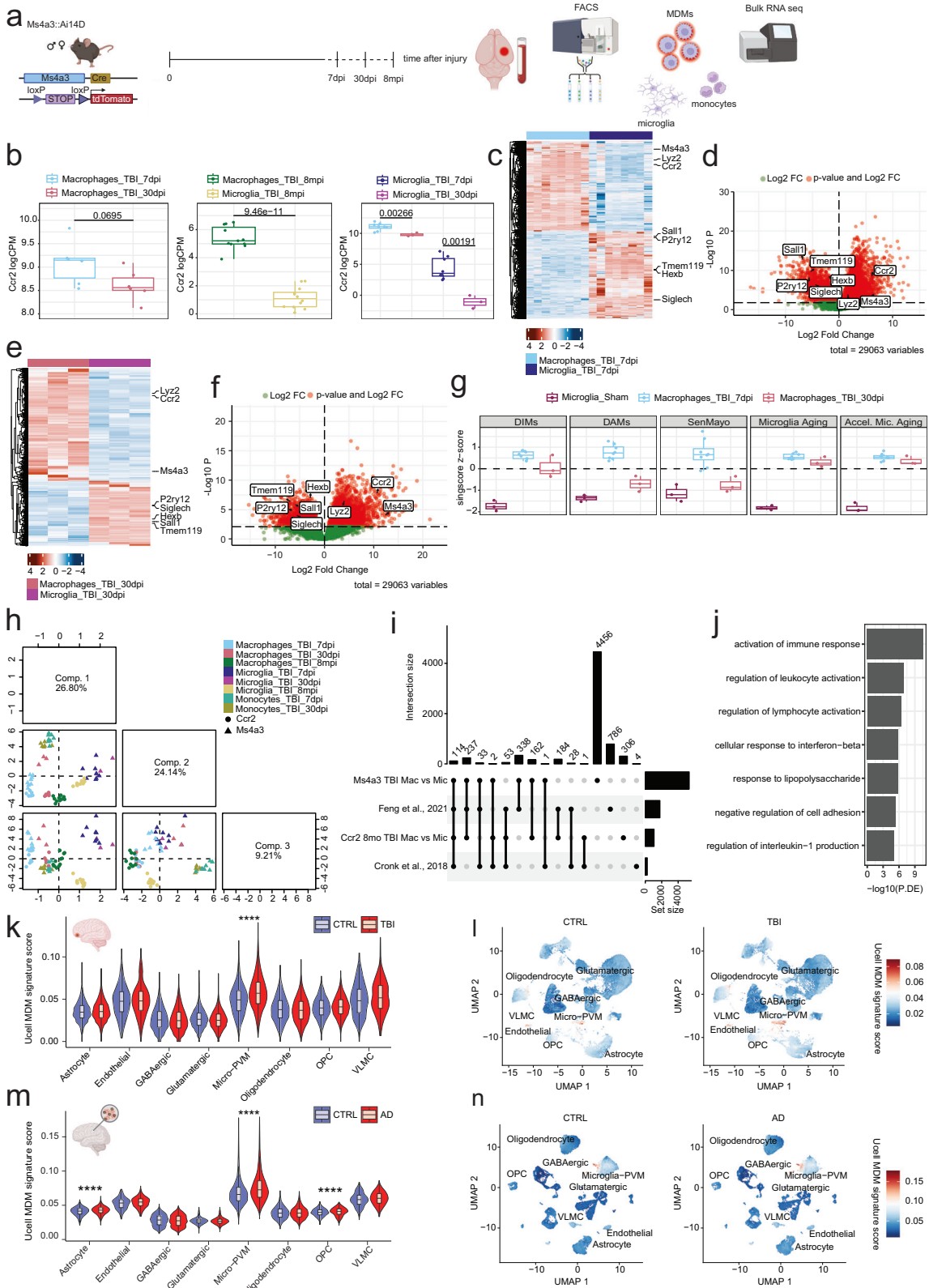

existing TBI studies[58], we hypothesize that a portion of infiltrated MDMs dies after the first wave of invasion, while a smaller niche (10%) durably persists in the TBI brain.

This durable population of MDMs tends to localize in the ipsilateral hippocampus, the thalamus underneath, and the choroid plexus, which is distinct from subacute time points when most of the infiltrated cells accumulate alongside the cavitation. We speculate that

after having fulfilled their acute function of debris clearance and tissue remodeling around the injured area, engrafted macrophages integrate into the network of resident macrophages. Supporting this hypothesis, we observed a reduction in cell volume at chronic time points compared to 7 days post-TBI. At 7 days, the larger cell volume likely reflects an enhanced phagocytic phenotype, corresponding to an increased demand for tissue clearance and repair subacutely after injury[61].

**Fig. 4 | Murine MDMs share a core transcriptomic signature enriched in human brain myeloid cells from TBI and AD patients. a** Experimental design - Ms4a3-cre::Ai14D mice were injured using the controlled cortical impact traumatic brain injury (TBI) model and brain samples were collected at different time points. Monocyte-derived macrophages (MDMs) (tdTomato + ), microglia (CD11b + , CD45$^{mid/low}$, tdTomato-) and inflammatory monocytes (Ly6G-,Ly6C$^{hi}$) were sorted from brain and blood by fluorescence-activated cell sorting (FACS) and processed for bulk RNA sequencing. **b** LogCPM of Ccr2 transcripts across datasets (two-sided Welch's t-tests with Holm multiple testing correction, *p-value* in the graph, left - Ccr2-creER$^{T2}$::Ai14D 30dpi vs 7dpi MDMs (df=8.994; 7dpi $n$ = 3 females, 3 males; 30dpi $n$ = 3 females, 3 males), middle - Ccr2-creER$^{T2}$::Ai14D 8 months post injury (mpi) MDMs vs microglia (df=20.341; MDMs: $n$ = 5 females, 6 males; Microglia: $n$ = 6 females, 6 males), right - Ms4a3-cre::Ai14D 30dpi vs 7dpi MDMs (df=8.228; 7dpi $n$ = 4 males, 4 males; 30dpi $n$ = 3 males) and microglia (df=6.475; 7dpi $n$ = 4 males, 4 males; 30dpi $n$ = 3 males)). **c** Heatmap of differentially expressed genes (DEGs) from Ms4a3-cre::Ai14D TBI MDMs and microglia 7 days after injury. **d** Volcano plot of DEGs from Ms4a3-cre::Ai14D TBI MDMs and microglia 7 days after injury. **e** Heatmap of DEGs from Ms4a3-cre::Ai14D TBI MDMs and microglia 30 days after injury. **f** Volcano plot of Ms4a3-cre::Ai14D DEGs from TBI MDMs and microglia 30 days after injury. **g** Singscore z-scores for reference datasets in Ms4a3-cre::Ai14D mice. **h** Projections of TBI samples from Ccr2-creER$^{T2}$::Ai14D and Ms4a3-cre::Ai14D models onto the first three components from sparse partial least squares

discriminant analysis (sPLS-DA) of singscore scores for cell type signatures from the Molecular Signatures Database (MSigDB). Individual animals are plotted (Ccr2-creER$^{T2}$::Ai14D mice (MDMs: 7dpi $n$ = 3 males, 3 females; 30dpi $n$ = 3 males, 3 females; 8mpi $n$ = 6 males, 5 females. Microglia 8mpi: $n$ = 6 males, 6 females) and Ms4a3-cre::Ai14D mice (MDMs: 7dpi $n$ = 4 males, 4 females; 30dpi $n$ = 3 males. TBI Microglia: 7dpi $n$ = 4 males, 4 females; 30dpi $n$ = 3 males. Sham Microglia: 7dpi $n$ = 2 males, 1 female. Monocytes: 7dpi $n$ = 4 males, 4 females; 30dpi $n$ = 3 males, 3 females)). Box plots depict data quartiles. (**i**) UpSet plot of the overlaps between DEGs from different datasets (see Methods). **j** Selected gene ontology (GO) terms enriched in the list of the 114 genes commonly enriched in MDMs. **k** Violin plots of Core MDMs UCell signature scores in human TBI single-nucleus RNAseq. **l** UMAP feature plot of Core MDMs UCell signature scores in human TBI single-nucleus RNAseq ($n$ = 7 TBI patients, $n$ = 7 controls). **m** Violin plots of Core MDMs UCell signature scores in Alzheimer's Disease (AD) dataset from Seattle Alzheimer's Disease Brain Cell Atlas (SEA-AD). **n** Umap feature plot of Core MDMs UCell signature scores in AD dataset from SEA-AD ($n$ = 42 AD patients with dementia, $n$ = 42 controls). ****$p$ < 0.0001 (one-sided Mann-Whitney U test). All $p$-values were corrected for multiple testing using Benjamini-Hochberg (FDR) correction. Only results with $p$ < 0.0001 are presented in the figure to emphasize the most robust findings. Source data are provided as a Source Data file. Created in BioRender. Frias, E. (2025) https://BioRender.com/bg3exxa.

Moreover, the % of tdTomato+ MDMs expressing the macrophage/microglia marker Iba1 increases over time after infiltration, reaching ~85% at 8 months. However, only ~10% of MDMs stained positive also for P2ry12, a well-established microglia-specific marker[62]. This suggests that MDMs do not fully differentiate into resident microglia, even long after engraftment. Our results are consistent with other fate mapping studies showing that MDMs do not contribute to the resident microglia pool during brain infection[63] and after microglia depletion[30,64]. Nevertheless, we show that MDMs are competent phagocytes both subacutely and chronically at 30dpi after TBI, in line with our previous findings in brain-engrafted macrophages (BEMs) that populate the brain after microglia depletion and have comparable phagocytic capacity[48]. Additional and conclusive proof that infiltrated macrophages maintain their distinct identity comes from bulk RNA sequencing data. By using two distinct yet complementary fate mapping systems (Ccr2-creER$^{T2}$ and Ms4a3-cre)[37,38], we showed the MDMs' transcriptomic profiles undergo dynamic changes after engraftment. Notably, even though MDMs at 8 months after TBI exhibit an intermediate transcriptome between macrophages and microglia at earlier time points, they still maintain their unique signature and downregulate microglia-specific genes. By comparing our data with reference datasets[41,52,53] we found that MDMs show a transcriptomic signature associated with disease, aging, and senescence. We propose that this signature may contribute to the persistent cognitive deficits observed following TBI, as we have previously shown that preventing the entry of MDMs into the brain can avert the development of cognitive impairments[16]. Finally, we generated a list of 114 core genes that are consistently shared across different published datasets focused on infiltrated macrophages[30,48] and have been previously implicated as drivers of critical processes, including neuroinflammation, cellular damage, and repair mechanisms in TBI (e.g., *Cybb*[65], *Mrc1*[66], *Tgfbi*[67], *Cd36*[68], *Mmp14*[69]). We observed a significant enrichment of this MDMs core signature in the brain myeloid population of a newly generated snRNAseq dataset derived from the orbitofrontal cortex of patients who died several years after TBI. Furthermore, this signature was also elevated in the brain myeloid cells of AD patients from the publicly available SEA-AD dataset. Given that TBI is a major environmental risk factor for dementia[70], these findings offer compelling evidence of the potential relevance of MDMs in the context of neurodegenerative diseases. Of note, 31 of these core genes were also found on the Druggable Genome (https://dgidb.org), a subset of the human genome that encodes for proteins capable of binding drug-like molecules (e.g., *Ca9, Anpep, Tlr8, Ahr, Mmp14,* and *Cxcr4*). Our findings not only will

facilitate the identification of these cells in future transcriptomic analyses but also provide valuable insights and promising targets for future studies aimed at developing therapeutic strategies targeting MDMs.

In conclusion, our study identifies a population of MDMs that persist in the TBI mouse brain for at least 8 months, exhibiting a transcriptomic signature linked to aging, senescence, and disease that is also found in human brain tissues. This study, however, has some limitations. First, the two fate-mapping models used in this study have inherent constraints. The Ccr2-CreERT2 system requires tamoxifen-induced activation, resulting in time-restricted labeling that may underrepresent the full population of MDMs infiltrating after TBI. In contrast, the Ms4a3-Cre constitutive model labels all MDMs, including both early and late infiltrating populations, but lacks temporal specificity, which limits insight into the dynamics of infiltration and engraftment. Similarly, the analyses performed across time points between the two fate-mapping models overlap only partially. Differences in labeling strategies between the two fate-mapping systems, partial overlap in analyses, and the lack of direct assessment of tdTomato labeling stability over extended time points should all be considered when interpreting the results. Moreover, the in vivo phagocytosis assay was performed only at 7 and 30 days post-TBI, providing useful information of subacute and early chronic MDM activity, but not direct insight into the functional state or contribution of MDMs at the 8-month post-infiltration time point. This study lacks additional validation through qPCR, in situ hybridization, or staining with standard lineage-specific markers such as CD206 and F4/80. Including these approaches in future work could strengthen the findings and provide valuable spatial context for MDM localization. Although TBI-induced cognitive deficits have been characterized using other behavioral tasks by our group and others[71–73] the present study focuses exclusively on evaluating learning and memory using the RAWM task. Finally, while previous studies from our lab and others have demonstrated a direct causal relationship between MDMs and TBI-induced cognitive deficits[14,16,18–23], we did not replicate these specific experiments in the current study. Instead, our work builds on the foundation of these earlier findings and takes a primarily descriptive approach, focusing on tracing the long-term trajectories and temporal transcriptomic dynamics of MDMs in TBI, a field that's unexplored and important for dementia. Future studies should leverage the findings presented in this study to investigate more precise modulation of MDM activity with the goal of developing targeted therapeutic strategies.

## Methods

### Animals

All experiments were conducted in accordance with National Institutes of Health (NIH) Guide for the Care and Use of Laboratory Animals and approved by the Institutional Animal Care and Use Committee (IACUC) of both the University of California, San Francisco (AN184326) and from Altos Labs, Inc (EB22-101-100) to ensure compliance with ethical standards and the humane treatment of animals.

Ccr2-creER[T2]-mKate2 (C57BL/6NTac background) mice were obtained from the University of Zurich (UZH). Ms4a3-cre mice (C57BL/6 J background) were obtained from the Singapore Immunology Network (SIgN), A*STAR. Ai14D reporter mice were purchased from Jackson Laboratory (#007914). Ccr2-creER[T2]-mKate2 and Ms4a3-cre mice were crossed with Ai14D mice to obtain Ccr2-creER[T2]::Ai14D and Ms4a3-cre::Ai14D mice. Female and male mice from both genotypes were 11-17 weeks of age at the time of surgeries for all experiments. Mice were group housed (by sexes and TBI state) in environmentally controlled conditions with a reverse light cycle (12:12 h light: dark cycle at $21 \pm 1°C$; ~50% humidity) and provided food and water ad libitum.

### Tamoxifen treatment and Traumatic Brain Injury surgery

Tamoxifen (Sigma, #T5648) was dissolved in corn oil and administered daily to Ccr2-creER[T2]::Ai14D mice by intraperitoneal injection of 5 mg/40 g body weight for 5 days[40]. To achieve the desired labeling of inflammatory monocytes (Ccr2 + ) in the blood, Ccr2-creER[T2]::Ai14D mice were treated with tamoxifen for 3 days before TBI, right after TBI and the day after, for a total of 5 injections.

Ccr2-creER[T2]::Ai14D and Ms4a3-cre::Ai14D mice were randomly assigned to each TBI or sham surgery group. Animals were anesthetized and maintained at 2–2.5% isoflurane. Controlled Cortical Impact (CCI) surgery was performed[19,44]. Briefly, mice were secured to a stereotaxic frame with nontraumatic ear bars. A midline incision exposed the skull, followed by a ~3.5-mm diameter craniectomy and removal of part of the skull, using a manual microdrill or a robot microdrill (Neurostar GmbH, customized). The coordinates of the craniectomy were anteroposterior, −2.0 mm, and mediolateral, + 2.0 mm with respect to bregma. Any animal that experienced excessive bleeding due to disruption of the dura was excluded from the study. After the craniectomy, the removed skull was discarded, and the contusion was induced using a 3 mm convex tip attached to an electromagnetic impactor (Leica). The contusion depth was set to 0.95 mm from dura with a velocity of 4.0 m/s sustained for 300 ms. Following the impact, the scalp was sutured. These TBI parameters were chosen to target, but not penetrate, the hippocampus. Sham animals underwent a similar procedure, but without the removal of the skull, and had no impact. Post-surgery, the mice recovered in a cage on top of a heated pad until they showed normal walking and grooming behavior, and then returned to their home cages. Only animals that fully recovered from the surgical procedures, as exhibited by normal behavior, healed sutures, and weight maintenance monitored throughout the duration of the experiments, were used.

### Radial arm water maze

The radial arm water maze (RAWM) task was used to test spatial learning and memory[44,45]. Pool dimensions were 118.5 cm in diameter with 8 arms, each 41 cm in length, and an escape platform. The escape platform was slightly submerged below the water level and water was rendered opaque by adding white paint (Crayola, #54-2128-053), so it was not visible to the animals. Visual cues were placed around the room. Two different versions of RAWM were used, a two-days version at 7 days post injury (dpi) and a three-days version at 30dpi and 8 months post TBI injury (mpi). In the first version, mice performed 9 trials on the learning day and 3 trials during the memory probe, 24 h after. In the second version, animals ran 6 trials/day during learning and 3 during the memory probe, one week later. On both versions, during both learning and memory days there was a 10 min inter-trial

interval. During a trial, animals were placed in a random arm (excluding the arm where the escape platform is located). Animals were allowed 1 min to locate the escape platform. On successfully finding the platform, animals remained there for 10 s before being returned to their warmed, holding cage. On a failed trial, animals were guided to the escape platform and then returned to their holding cage 10 s later. The escape platform location was the same in each experimental cohort, whereas the start arm varied between trials. Mice behavioral videos were acquired using Ethovision XT (Noldus Information Technology). The number of errors from 3 consecutive trials was averaged in one block. Data was graphed as the number of non-targeted entries (number of errors).

### Flow Cytometry

**Blood.** To test the efficiency of tamoxifen-induced labeling of peripheral Ccr2+ monocytes, blood was collected from Ccr2-creER[T2]::Ai14D via tail vein puncture. A small nick was made in the tail vein using a scalpel, and ~40 μl of blood was removed using a pipette and placed in a small tube containing EDTA (Sigma, #E8008). In Ms4a3-cre::Ai14D mice, ~200 μl of blood was extracted by cardiac puncture once completely anesthetized before tissue collection. Red blood cells were lysate using RBC lysis buffer (BioLegend, #420301) and samples were then blocked with CD16/32 Fc block (BD Biosciences #553141) and stained with fluorophore-conjugated antibodies at 1:100 dilution: CD11b-AF700 (BD Pharmingen, #557690), CD11b-BV421 (BD Pharmingen, #562605), CD45-FITC (BD Pharmingen, #553080), CD45-APC-Cy7 (BioLegend, #304014), Ly6C-V450 (BD Pharmingen, #560594), Ly6C-APC (BD Pharmingen, #560600) Ly6G-PE-Cy7 (BD Pharmingen,# 560601) and Ly6G-BV711 (BD Pharmingen, #563979). Cells were then washed in FACS buffer (1×DPBS with 0.5% BSA fraction V and 2% FBS) and used for analyses. Data were collected on BD FACSymphony™ A3 Cell Analyzer and on FACSAria™ (III and Fusion) cell sorters (BD Biosciences, V8.0.1), and analyzed with FlowJo software (FlowJo, LLC).

**Brain.** Mice were euthanized via a lethal overdose of ketamine/xylazine or by $CO_2$ inhalation and perfused with cold PBS. Brains were immediately removed and dissociated using a Neural Tissue Dissociation kit (P) (Miltenyi Biotec, #130-092-628). Cells were then resuspended in 30% Percoll solution diluted in RPMI medium, and centrifuged at 800 g for 20 min at 4 °C. Cell pellets were washed with FACS buffer (1×DPBS with 0.5% BSA fraction V and 2% FBS or Rockland, #MB-086-0500), blocked with mouse CD16/32 Fc block (BD Biosciences, #553141) and then stained with fluorophore-conjugated antibodies at 1:100 dilution: CD11b-AF700 (BD Pharmingen, #557690), CD11b-BV421 (BD Pharmingen, #562605), CD45-FITC (BD Pharmingen, #553080), CD45-APC-Cy7 (BioLegend, #304014), Ly6C-V450 (BD Pharmingen, #560594), Ly6C-APC (BD Pharmingen, #560600) Ly6G-PE-Cy7 (BD Pharmingen,# 560601) and Ly6G-BV711 (BD Pharmingen, #563979). Cells were then washed in FACS buffer and used for analyses or sorted. Data were collected on BD FACSymphony™ A3 Cell Analyzer and on BD FACSAria™ (III and Fusion) cell sorters (BD Biosciences, V8.0.1), and analyzed with FlowJo software (FlowJo, LLC).

### Immunofluorescence

For immunohistochemistry analysis, animals were lethally overdosed with ketamine/xylazine, followed by PBS perfusion. Brains were fixed in ice-cold 4% paraformaldehyde, pH 7.5 (PFA, Sigma Aldrich, St. Louis, MO, 441244) overnight, cryo-protected in 15% and 30% sucrose (Fisher Scientific), and sectioned into 20 μm coronal slices using a Leica cryostat (Leica Microsystems). Slides were then placed in 100% methanol at −20 °C for 10 minutes, blocked in BSA 5% in TBS-Tween, and then stained with antibodies (488-conjugated rabbit anti-Iba1, Cell Signaling, #20825, 1:100, and anti-P2ry12, Anaspec, #AS-55043A, 1:400) overnight at 4 °C and then 2 h at room temperature with secondary antibodies. Nuclei were stained with DAPI. Tissues were fixed

using ProLong Gold (Invitrogen, #P36930) and a standard slide cover sealed with nail polish.

Pericontusional regions (top quarter of a coronal brain section) were analyzed by acquiring z-stacked tiled images on a Axioscan 7 slide scanner (Zeiss) using a 20x lens. For each mouse, cavitation images were acquired at 3 different coordinates from rostral to caudal: approximately Bregma −1.34 mm (Anterior), −1.81 mm (Central), and −2.54 mm (Posterior).

Imaris (version 10.2.0) is an interactive software used to visualize, analyze, and quantify 3D microscopy data[74,75]. It uses deep learning algorithms to detect and segment objects within images, which helps improve segmentation accuracy and provide in-depth insights into cellular structures and processes as compared to traditional methods. We used its machine learning-based segmentation to create surface renderings of Iba1, P2ry12, DAPI staining, and tdTomato signal. We exported the parameters Volume to measure the total three-dimensional space occupied by an object, and Overlapped Volume Ratio to Surfaces to quantify the extent of overlap between two or more surfaces. A tdTomato/DAPI overlapped volume ratio of 0.5 was used as the threshold criterion to include cells in the analysis.

### In vivo phagocytic assay

To test the phagocytosis capacity of infiltrated macrophages, 2 µl of 2µm latex beads (Sigma Aldrich, #L0280) diluted 1:10 in saline were injected into the right hippocampus (ipsilateral to the TBI) using the following coordinates: bregma, AP−1.6 mm, ML + 1.6 mm, and DV −2 mm. Mice were euthanized by ketamine/xylazine overdose three days post-injection and phagocytosis was evaluated by immuno-fluorescent staining (colocalization of blue beads in tdTomato+ (Ccr2+) cells). Nuclei were stained with H3K9me3 antibody (abcam, #ab176916) and visualized in green, as the blue channel was used by the latex beads. 40x Z-stack tiled images of the injection site (top half of the TBI hemibrain) were acquired on a Axioscan 7 slide scanner (Zeiss). 2/4 images were acquired for each animal. One representative image was acquired using an LSM980-Airyscan2 (Zeiss) laser scanning confocal. Maximum Intensity Projection and normalization to the 1st and 99th percentile was performed for each channel of the z-stack. tdTomato+ macrophages and blue beads were automatically segmented using the 2-class K-means algorithm implemented in MATLAB. Circularity index was computed as $(4*pi*Area/Perimeter\textasciicircum 2)*(1 - 0.5/r)\textasciicircum 2$ where $r = Perimeter/(2*pi) + 0.5$, for each segmented connected component in the tdTomato channel. Blobs with circularity index <0.7 were excluded by the analysis, as well as cells with cross-sectional area outside the range of 5-20 µm. Intersection between each remaining Ccr2+ cell and the blue beads segmentation mask was computed, and each cell with non-zero intersection was counted as positive in blue beads and tdTomato+ cell colocalization. Data are graphed as % of tdTomato+ macrophages engulfing one or more beads (number of macrophages that colocalize with beads in the acquired image/ total number of macrophages in the image*100).

### Bulk RNA sequencing

**Library preparation.** CD11b + CD45+ tdTomato+ cells (MDMs) were sorted from whole brains of Ccr2-creERT2::Ai14D mice at 7 days, 30 days, and 8 months after TBI and from Ms4a3-cre::Ai14D mice at one week and one month after TBI. For the first two time points (7 and 30 days, Ccr2-creERT2::Ai14D mice), samples were processed as follows. cDNAs were generated using the SMART-Seq v4 Ultra Low input RNA kit for sequencing (Takara Bio, 634891), according to the vendor protocol with 14 cycles of cDNA amplification. At the last step, the cDNA was eluted in 30ul of RNase/DNase-free water. Subsequently, 30ul of cDNA was used as a starting material with Illumina DNA Prep library kit (Illumina, 20018705) according to the vendor protocol, with final library PCR amplification of 10 cycles. After library completion, individual libraries were pooled equally by volume and quantified on

the Fragment Analyzer (Agilent, DNF-474). Quantified library pool was diluted to 1 nM and sequenced on MiniSeq (Illumina, FC-420-1001) to check for quality of reads. Finally, individual libraries were normalized according to MiniSeq output reads, specifically by % protein coding genes, and were sequenced on two lanes of HiSeq4000 SE50 for a total of ~30 M reads per sample.

For the 8mpi time point (Ccr2-creER^T2::Ai14D mice) and the Ms4a3-cre::Ai14D mice, cDNAs were generated using the SMART-Seq® mRNA LP (with UMIs) kit for sequencing (Takara bio, 634765), according to the vendor protocol. The cDNA amplification cycle was adjusted to 19 cycles. PCR-amplified cDNA was purified by immobilization on NucleoMag NGS Clean-up and Size Select beads. The beads were then washed with 80% ethanol, and cDNA is eluted with ~15ul of Elution Buffer. Subsequently, 8 ul of cDNA 0.125 ng/µL was used as a starting material for library preparation using the Unique Dual Index (UDI) kits (Takara Bio, 634752-5) according to the vendor protocol, with final library PCR amplification of 14 cycles. The PCR-amplified library was purified individually by immobilization on NucleoMag NGS Clean-up and Size Select beads. The beads were then washed with 80% ethanol, and the library was eluted with nuclease-free water.

Quality control and quantification were performed using the Qubit dsDNA HS Assay Kit (Thermofisher, 32854) and TapeStation D5000 Reagents (Agilent, 5067-5589) and Screentape (Agilent, 5067-5588). Samples were normalized and pooled individually for sequencing. An additional double-sided SPRI selection was performed on the pool prior to sequencing. Libraries were sequenced using the NextSeq 2000 P3, 300 cycles kit (Illumina, 20040561).

**Pre-processing and QC.** RNA-seq FASTQ files were trimmed, aligned to the mouse reference genome (GRCm39, v108), and quantified using the nf-core/rnaseq pipeline with the star_salmon option (v3.12.0)[76]. Gene-level length-scaled counts were used for all downstream analyses. In the Ms4a3-cre::Ai14D and 8-month Ccr2-creER^T2::Ai14D datasets, UMIs were trimmed from all reads before alignment and quantification, and technical replicates from different wells were summed together. GC-content correction offsets were calculated for the Ms4a3 datasets using the cqn package[77] and latent batch effects were removed using Surrogate Variable Analysis (SVA)[78].

**Differential gene expression.** Differentially expressed genes (DEG) between conditions at each time point were identified within each dataset using the quasi-likelihood framework implemented in edgeR. Raw length-scaled counts were filtered to retain genes with a minimum of 10 counts within biological groups and trimmed-mean of M-value (TMM) normalization factors were calculated. Empirical Bayes estimates of the quasi-likelihood dispersions were robustified against outlier genes. Thresholds of adjusted $p$-value < 0.05 and absolute log-fold-change > 0 were used to identify differentially expressed genes. Differential expression testing was not performed between datasets (Ccr2 7dpi & 30dpi, Ccr2 8mpi, Ms4a3) due to batch confounding.

**Single-sample scoring (singscore).** To compare groups across batch-confounded datasets, the filtered counts tables from all datasets were merged using the intersection of expressed genes. Genes were then ranked by expression within each sample and independently scored for enrichment against published transcriptomic signatures and gene sets from several databases (i.e., Reactome[49], ImmuneSigDB and Cell Type Signature Signatures from MSigDB[51,79,80]) using the singscore package[50]. The Reactome terms were downloaded from https://www.reactome.org. The ImmuneSigDB and Cell Type Signatures were collected from the msigdbr package. Enrichment scores were then compared between groups and datasets. The DIMs and DAMs signatures were collected from Supplementary Table 1 of Silvin et al. [41]. The SenMayo signature was collected from Supplementary Data 1 of Saul et al. [53]. The Microglia Aging and Accelerated (Accel.) Microglia Aging

signatures were collected from Holtman et al. [54] as stored in the 2021 HDSigDB database at https://www.maayanlab.cloud.

**Sparse partial least squares discriminant analysis (sPLS-DA).** First, PLS-DA was performed using the mixOmics package[81] for Singscore scores from each database (i.e., Reactome, MSigDB cell type signatures, ImmuneSigDB) using at least 10 components. Model performance was assessed using M-fold cross-validation with the "perf" command with 50 repetitions. 5 folds were used for the Ccr2 datasets, and 3 folds were used for the combined analysis of the Ccr2 and Ms4a3 datasets. Next, sparse PLS-DA (sPLS-DA) was performed using one additional component than the optimal number of components identified from PLS-DA. Cross-fold validation was performed to determine the optimal number of components and important features per component with the same number of folds and repetitions. The resulting number of components and features was used to train the final sPLS-DA model. 3 components were selected for the Ccr2 analysis, and 9 components were selected for the combined analysis of Ccr2 and Ms4a3 with the cell type signatures database. The selected features for each component were visualized using heatmaps of singscore z-scores for all terms with non-zero contributions to any component.

**Overlap with the core MDMs signature.** MDMs signatures relative to resident microglia from our Ccr2 and Ms4a3 datasets were overlapped with signatures collected from external datasets. "Ms4a3 TBI Mac vs Mic" refers to the union of genes that are upregulated in TBI MDMs relative to TBI MDMs in the Ms4a3 model across 7dpi and 30dpi. "Feng et al., 2021" refers to genes that are upregulated in brain-engrafted macrophages (BEMs) collected from irradiated mice treated with PLX versus microglia collected from control mice without PLX. "Ccr2 8mpi TBI Mac vs Mic" refers to genes upregulated in 8-month TBI MDMs versus 8-month TBI microglia. "Cronk et al., 2018" refers to the intersection of genes that are upregulated in brain-engrafted macrophages versus microglia across three models: Cx3cr1^CreER/+^::Csf1r^Flox/Flox^ followed by bone marrow transplantation (BMT), traditional irradiation and BMT, and BMT with PLX5622 treatment. UpSet plots were also created using the ComplexHeatmap package.

The 114 overlapping MDMs signature genes were annotated to Gene Ontology (BP) terms using the goana function in limma. The universe was set as genes that were commonly expressed across all of the examined datasets (14,825 genes).

**Heatmaps and volcano plots.** Heatmaps were created using the ComplexHeatmap package[82] with z-scores of logCPM values or singscore scores. Volcano plots were created using the EnhancedVolcano package.

**Human Samples**
Brain donation studies at the University of Washington (UW) were approved by the UW School of Medicine Compliance office and Institutional Review Board (IRB), and all brain donation was collected with informed consent. In 2016, the UW IRB issued an official determination that our repository work does not meet the metric of human subject's research as we are collecting samples from deceased individuals. Our practices are now informed by the US Revised Uniform Anatomical Gift ACT 2006 (Last Revised or Amended in 2009) and Washington Statute Chapter 68.64 RCW. Brain donors in this study were part of the Pacific Northwest Brain Donor Network, which collects tissue from military veterans and civilians, both with and without a history of traumatic brain injury in collaboration with local Seattle area medical examiners. Consent for brain donation was obtained from the legal next of kin who agreed to be contacted and were interested in the UW research program. History of TBI ("Did they/the donor ever have a head injury?") and other

metrics were obtained via family questionnaire. Data collected included mechanisms of TBI, loss of consciousness, symptoms after the injury, and medical and psychiatric conditions. Control cases never had a known head injury or other history of neurologic disease and were age- and sex- matched to the TBI cases ($n = 7$). TBI cases were chosen that had available frozen tissue and a head injury associated with hospitalization, the majority of which were noted to be associated with a coma by family members ($n = 7$). Every case underwent a neuropathologic evaluation that included full assessment for neurodegenerative disease by current consensus criteria including Alzheimer's disease, Lewy body disease, TDP-43 pathologies, chronic traumatic encephalopathy (CTE), and other tauopathies as described in Latimer et al.[83] Over 17 cortical regions with numerous sulci sampled were stained with phosphorylated tau to evaluate for CTE pathology. Bilateral orbitofrontal and anterior temporal lobes were also always evaluated to assess for contusion in every case. For this study, the orbitofrontal cortex was chosen for microscopic evaluation, given its frequent vulnerability in TBI[84,85]. A summary of the clinical characteristics and neuropathology is given in Supplementary Table S1.

**Single nuclei RNA sequencing**
**Library preparation.** Frozen human brain tissues were homogenized in a Dounce homogenizer using Lysis Buffer (10 mM Tris-HCl, pH 7.4, 10 mM NaCl, 3 mM MgCl$_2$, and 0.025% NP-40) and incubated on ice for 15 minutes. The homogenate was filtered through a 30-µm mesh to remove debris and centrifuged at 500 g for 5 minutes at 4 °C to pellet the nuclei. Nuclei were then washed and filtered twice using Nuclei Wash Buffer (1% BSA in PBS with 0.2 U/µl RNasin, Promega). For myelin debris removal, the nuclei were resuspended in 500 µl of Nuclei Wash Buffer, mixed with 900 µl of 1.8 M sucrose, and carefully layered over 500 µl of 1.8 M sucrose. This gradient was centrifuged at 13,000 g for 45 minutes at 4 °C. The isolated nuclei were then resuspended in Nuclei Wash Buffer at a final concentration of 1000 nuclei/µl and passed through a 40-µm FlowMi Cell Strainer for further purification. Isolated human nuclei were processed using the Chromium Single Cell 5′ Reagent Kits (10x Genomics) for droplet-based 5′ end massively parallel single-cell RNA sequencing. Sequencing libraries were generated and run on Illumina sequencers at the McDonnell Genome Institute. Sample demultiplexing, barcode processing, and single-cell counting were performed using the Cell Ranger Single-Cell Software Suite (10x Genomics). The cellranger count command was used to align reads to the human reference genome (GRCh38).

**Pre-processing and QC.** Raw sequencing reads were aligned to the GRCh38 human reference genome using the Cell Ranger pipeline (v.8.0.0) to generate the raw RNA count matrix. The resulting count matrix was further processed using the Seurat R package[86] (v5.2.0) for downstream quality control, normalization, and dimensionality reduction. To ensure high data quality, filtering was performed using PopsicleR[87] (v0.2.1) with strict thresholds. The criteria included: (1) A minimum of 500 and a maximum of 12,000 detected genes per cell (G_RNA_low=500, G_RNA_hi=12,000). (2) The number of unique molecules per cell was required to fall within a reasonable range (U_RNA_low=1500, U_RNA_hi = 37,000). (3) Cells exceeding the upper limits for mitochondrial, ribosomal, and dissociation gene percentages (percent_mt_hi=5, percent_ribo_hi=100, percent_disso_hi=100) were removed. After filtering, 98,492 cells met the inclusion criteria for analysis. The dataset had an average read depth of 9,226 unique RNA molecules per cell, with an average of 3312 unique genes detected per cell. Doublets were detected and removed using the CalculateDoublets function within PopsicleR. RNA normalization was performed using the log-normalization method within Seurat's NormalizeData function, followed by scaling using ScaleData. This was

followed by Principal Component Analysis (PCA), on the top 2,000 highly variable features, retaining the first 30 principal components for downstream clustering and visualization. For cell type annotation, we utilized Azimuth[57] (v0.5.0) for reference-based mapping, using the humancortexref reference dataset to assign cell identities. The classification was performed at two hierarchical levels: Level 1: Broad categorization into GABAergic neurons, Glutamatergic neurons, and non-neuronal populations. Level 2: Further refinement into specific neuronal and non-neuronal subtypes.

**MDMs core signature analysis.** Next, we applied UCell[88] (v2.7.7) to compute Mann-Whitney U statistic-based signature scores for the human orthologues of our MDM signature genes (84 genes, Supplementary Data 1) across all cells. To evaluate differences between Control and TBI cells within each Level 2 (L2) cell annotation, we performed Mann-Whitney U tests (one-sided, TBI greater than Control) using the stat_compare_means function. Similarly, we applied UCell to a downsampled subset of the Seattle Alzheimer's Disease Brain Cell Atlas (SEA-AD[42]), reducing the dataset to 50,000 cells using Seurat's representative sketching method. We then compared Normal and diseased cells using Mann-Whitney U tests (one-sided, Diseased greater than Control), again leveraging the stat_compare_means function. All *p-values* were corrected for multiple testing using Benjamini-Hochberg (FDR) correction. Only results with $p < 0.0001$ are presented in the figure to emphasize the most robust findings.

**ShinyCell interactive visualization.** We aimed to make our single-cell human TBI single-cell dataset accessible as a resource. We utilized the ShinyCell package[89] (v2.7.7) as the foundation for developing our interactive web application. This application was deployed to the rsconnect server and is available at the following link: https://altoslabs.shinyapps.io/tbi-human-atlas/.

### Statistics and reproducibility

All data were analyzed with GraphPad Prism 9 statistical software. Cognitive performance in the RAWM maze was analyzed as a Two-way repeated measure (RM) analysis of variance (ANOVA) (covariates: TBI and time) followed when appropriate by Šidák multiple comparisons test. Differences in phagocytosis were analyzed using a two-sided unpaired Student's *t*-test. Cell volume data across time points were analyzed using one-way ANOVA followed by Tukey post hoc test or two-sided unpaired Student's *t*-test. Group outliers were determined (ROUT method, $Q = 1\%$) and excluded from analysis. P values below 0.05 were considered significant. Individual animal scores represented by dots, lines depict the data mean and SEM. Comparisons between the expression of specific genes were visualized using boxplots and tested using two-sided Welch's t-tests with Holm multiple testing correction. Differences in MDMs signature enrichment in human samples were tested using the stat_compare_means function (one-sided Mann-Whitney U test). Individual statistical analysis is denoted in the figure legends. All data analyses were conducted with investigators blinded to the experimental group assignments. Boxplots of scores and expression values are defined by the median and interquartile range, with whiskers identifying outliers defined by 1.5*IQR.

Figure 2b, c, i present representative images from experiments corresponding to the analyses shown in Figs. 2d–g, k, respectively. These data were obtained from a single experimental cohort, with multiple biological replicates. Sample size is indicated in the figure legends.

Software used to analyze the bulk and single-nucleus RNA-sequencing data are provided in Supplementary Table S2.

### Reporting summary

Further information on research design is available in the Nature Portfolio Reporting Summary linked to this article.

## Data availability

The raw files for the datasets generated in this manuscript have been deposited in the National Center for Biotechnology Information Gene Expression Omnibus (GEO). Bulk RNA-seq libraries of CD11b + CD45+ cells from whole brains of Ccr2-creER$^{T2}$::Ai14D (7, 30 days after TBI) have been deposited under accession code: GSE283558. Bulk RNA-seq libraries of CD11b + CD45+ cells from whole brains of Ccr2-creER$^{T2}$::Ai14D (8 months after TBI) have been deposited under accession code: GSE283556. Bulk RNA-seq libraries of CD11b + CD45+ cells from whole brains of Ms4a3-cre::Ai14D have been deposited under accession code: GSE283560. The single-cell RNA-seq libraries of human TBI samples have been deposited under accession code: GSE294775. An interactive shiny app for visualizing the TBI datasets is available here: https://altoslabs.shinyapps.io/tbi-human-atlas/. The Seattle Alzheimer's Disease Brain Cell Atlas is publicly available and was downloaded from: https://portal.brain-map.org/explore/seattle-alzheimers-disease/seattle-alzheimers-disease-brain-cell-atlas-download?edit&language=en. The Druggable Genome database was accessed through: https://dgidb.org/. All data supporting the findings of this study are available within the paper and its Supplementary Information. Source data are provided with this paper.

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

## Acknowledgements

We thank Professor Burkhard Becher (University of Zurich) for providing Ccr2-creERT2-mKate2 mice and Professor Florent Ginhoux (Singapore Immunology Network (SIgN), A*STAR) for providing Ms4a3-cre mice. We thank the Genomics CoLabs of the University of San Francisco, California (UCSF) that run part of the Bulk RNA sequencing analysis. We thank Eric Griffis (Altos Labs) for helping with the acquisition of images for the phagocytosis assay, Yuewen Zheng (Altos Labs) for her contribution to the sequencing analysis, Aniket Tolpadi (Altos Labs) for assisting with processing Imaris raw data, Shruti Suresh (Altos Labs), Ariana Wei (Altos Labs), Valentina Frattini (UCSF), Kim Chung (UCSF) for helping with sample processing. Human brain donation and neuropathological analysis at the University of Washington was supported by the following grants: U24 AG072458, DoD W81XWH-21-S-TBIPH2, U24 NS135561, U01 NS13748, RF1NS115268, U01NS086625, P30AG066509, P50AG005136, U01AG006781, U19AG066567, U19AG060909. We also thank the donors and their families for their contribution to science. Moreover, we thank Siling Du (WashU) and Yingyue Zhou (WashU), for processing the human brain samples for single-nucleotide RNA sequencing. Images were created with BioRender.com.

## Author contributions

Conceptualization: M.S.P., B.A.Y., S.R., X.F., K.K., K.A.T., A.L.N., M.C. Experimental work: M.S.P., W.L., R.S. Data analysis: M.S.P., B.A.Y., K.A.T., E.S.F., V.P., with the support of M.M., S.T. Manuscript writing: M.S.P., B.A.Y., S.R. All the authors made contributions to the manuscript editing.

## Competing interests

The authors declare no competing interests.
