## [Transparent Peer Review file · Nature Communications]

Fate mapping of peripherally-derived macrophages after Traumatic Brain Injury in mice reveals a long-lasting population with a distinct transcriptomic signature that is also enriched in human brain myeloid cells

Corresponding Author: Professor Susanna Rosi

Version 0:

Reviewer comments:

Reviewer #1

(Remarks to the Author)

This manuscript, "Fate mapping of peripherally derived macrophages reveals a long-lasting engrafted population that maintains a distinct transcriptomic profile for up to 8 months after Traumatic Brain Injury" by Paladini et al., reveals the long-term persistence and functional role of monocyte-derived macrophages (MDMs) in the brain following traumatic brain injury (TBI). Using innovative fate-mapping models, the authors show that MDMs remain distinct from microglia, retain phagocytic activity, and exhibit aging- and disease-related signatures up to 8 months post-TBI. The work addresses a critical gap in understanding MDMs' contribution to chronic neuroinflammation and cognitive deficits, providing robust evidence and potential therapeutic targets.

General Comments:

The existing literature extensively documents the neurodegenerative and detrimental roles of MDMs in the neuroinflammatory response following TBI. The macrophage and microglial cell signatures post-TBI are well-established. While characterizing this response reinforces previous findings, it offers only descriptive insights without shedding light on underlying mechanisms. Furthermore, the characterization does not convincingly demonstrate a causative link to the development of memory and learning deficits.

In addition, the study's focus on MDMs in injured brains at this stage fails to provide meaningful data regarding their phagocytic or alternative functional roles. The lack of functional modulation or intervention experiments leaves it unclear whether the observed characterization results from MDM activity or reflects a chronic inflammatory environment. As such, the work does not advance understanding of the peripheral immune response to TBI or elucidate novel functional insights about MDMs. Given the extensive preclinical studies in the literature that have characterized MDMs and explored various treatments, this study does not contribute significantly novel information. The reviewer recommends incorporating studies that evaluate the role of MDMs in the context of TBI. Integrative approaches, particularly those that address current knowledge gaps and provide robust functional outcomes, would strengthen the conclusions and enhance the study's overall impact.

Major Comments:

Abstract:

The abstract should not include general background information on the topic, as this is already well-established in the field. Instead, it should focus on the specific contributions this study makes to advancing knowledge about MDMs in the context of TBI.

Introduction:

While the role of MDMs in TBI is well-documented, it is also worth noting that nanoparticle-based drug delivery systems and vagus nerve stimulation are being investigated to modulate MDM activity, resolve neuroinflammation, and restore cognitive function post-TBI. However, this study does not explore or analyze the functional roles of MDMs following TBI, which limits its scope and impact.

Methods:

- The study's reliance on the *Ccr2-creERT2* model with limited validation through complementary models (e.g., *Ms4a3-cre*) raises concerns about the generalizability of the findings. Furthermore, no clear justification for selecting these specific mouse models is provided, leaving the rationale unclear.
- While the *Ccr2-creERT2::Ai14D* system achieves high initial labeling efficiency (~80–85%), the rapid turnover of Ly6Chi monocytes and the decline in labeled cells over time (~25% after one week) may lead to underrepresentation or bias in longitudinal tracking of MDMs.
- The study misses an opportunity to utilize cellular markers that accurately distinguish the phenotypes of microglia and macrophages, leading to potentially invalid interpretations. Key markers such as CD68 or CD206 for microglia and F4/80 for peripheral macrophages should have been used. Additionally, the immunofluorescence data should be reanalyzed to improve accuracy and clarity.
- The assertion that MDMs transition to a phenotype distinct from microglia is insufficiently supported. While markers like *Iba1* and *P2ry12* suggest partial differentiation, the absence of statistically significant increases in *P2ry12*+ expression over time undermines the claim of meaningful microglia-like adaptation. For instance, claiming that *Iba1*, expressed by macrophages, is also expressed by microglial cells (resident cells in most cases) lacks sufficient validation.
- The legends and results do not specify the number of animals and stats analyzed per group, the regions assessed in each animal, or the statistical methods employed. These omissions reduce the study's transparency and reproducibility.

Results:

- Extensive research has established that CCI can result in long-term cognitive deficits in mice, persisting for several months post-injury. However, the added value of a descriptive characterization at 8 months post-TBI is unclear. The mere persistence of MDMs up to 8 months post-TBI is insufficient to establish their functional relevance. Persistence alone does not imply a pathological role unless causality is definitively demonstrated. A more informative approach could involve analyzing the inflammatory response through neuroimaging at multiple time points post-TBI, extending up to 2 years.
- Neuroimaging techniques should be employed to accurately characterize the progression of "Disease Inflammatory Macrophages" (DIM) at a chronic stage. Live imaging via two-photon microscopy to monitor macrophage infiltration and activity in the injured brain or combining imaging with systemic administration of fluorescently labeled antibodies to detect activation markers would provide more robust and functional insights. The enrichment of DIM and disease-associated microglia (DAM) signatures may indicate overlap; however, it risks confounding the interpretation of MDM-specific contributions, especially without proper functional dissection. The current characterization of microglia/macrophage populations is insufficiently precise and leads to potentially erroneous conclusions.
- Transcriptomic profiling does reveal distinct temporal signatures, but the overlap with aging, senescence, and disease-associated profiles may reflect a generalized response to chronic inflammation rather than effects specific to TBI.
- The statement, "Macrophages are professional phagocytes," is an intriguing observation but lacks scientific depth or relevance in the context of this study. It fails to add meaningful insight.
- The claim regarding "morphological changes associated with phagocytic marker *Iba1* over time after engraftment, not fully differentiated into microglia" is problematic. Without proper characterization of the cellular phenotype, such assertions lack validity. Morphological changes, such as reduced cell volume, are more likely reflective of survival adaptations rather than functional differentiation or pathological significance.
- Genes expressed by activated microglia or peripherally derived macrophages should be analyzed to validate RNA-seq results. Characterization of pro-inflammatory cytokines, such as IL-1 β , TNF- α , and IL-6, co-localized with *Iba1* (but not F4/80), would help distinguish pro-inflammatory microglia from macrophages.
- Identifying 29 core genes shared by MDMs across models is intriguing but remains incomplete without functional validation. Whether these genes are causal in cognitive deficits or merely epiphenomenal to the injury response is unclear.
- Pharmacological studies should be integrated to strengthen these findings, including the depletion of macrophages using liposome-encapsulated clodronate or CCR2 inhibitors. Then, the impact on TBI outcomes should be assessed. This approach would provide a more definitive understanding of macrophage contributions than essential gene signature characterization.
- Neutrophil populations infiltrating the brain must also be evaluated to determine their short-term, long-term, and beneficial or detrimental roles in TBI, particularly in macrophage recruitment and activity.
- To elucidate the function of MDMs, the immune response pathways implicated in TBI should be validated by qPCR, ELISA, immunostaining, and other methods. Special attention should be given to genes linked to aging and senescence that are upregulated in MDMs.
- Although the study highlights long-term MDM persistence and transcriptomic changes, it does not address whether modulating MDM activity or recruitment could mitigate cognitive impairments. This omission represents a critical gap in translational relevance.
- Behavioral Testing: The study proposes a link between cognitive deficits and MDMs but lacks direct mechanistic evidence connecting MDM-specific transcriptomic changes to cognitive impairments.
- "Cognitive impairments in RAWM": While the radial arm water maze (RAWM) assesses spatial memory, additional cognitive domains such as executive function and attention remain untested. Exploring these domains could offer a more comprehensive understanding of TBI-induced impairments.
- The absence of comparisons with other TBI models or cognitive assays limits the generalizability of the findings. Incorporating other paradigms, such as the Morris Water Maze (MWM), novel object recognition, or Barnes Maze, would strengthen the study's conclusions.

Figures

The figures should be reconstructed to enhance clarity, reduce redundancy, and effectively highlight the most significant

findings. The font size and image quality should also be improved to ensure legibility. Currently, several errors and inconsistencies undermine the impact of the figures:

1. Figure 1c: The reliance on tamoxifen-induced labeling may not accurately reflect newly recruited or replenished MDM populations, especially at 30 days or 8 months post-TBI. This limitation should be addressed or acknowledged in the figure's interpretation.
2. Figure 2a: The brain map shows identical representations for the anterior and central areas, which introduces ambiguity and requires correction.
3. Figure 2b: The representative images of the damaged cortical region do not correspond to the lesion measurements presented in Figure 2f. This discrepancy should be resolved to ensure consistency between visual and quantitative data.
4. Individualized Data Points: All figures should include markers to indicate data from individual samples, enhancing transparency and allowing for a more robust interpretation of variability.
5. Figure 2: The volumetric density (μm^3) measurement of tdTomato+Iba1+ cells across three selected coronal sections is overly simplistic and inappropriate for drawing conclusions about microglial/macrophage cell density. Instead, the following methods should be employed:
 - o Stereological Analysis: Perform stereological assessments in the three proposed regions to accurately quantify cell density.
 - o Morphological Analysis: Sholl analysis and NeuroLucida can be used to evaluate how microglial/macrophage morphology changes over time and across regions.

Discussion

To contextualize this study's findings, it is necessary to thoroughly discuss the previous literature on central and peripheral inflammation following TBI in both preclinical and clinical studies. This would help clarify what the study uniquely contributes to understanding the causes of behavioral deficits, particularly cognitive impairments. While the study asserts that preventing MDM infiltration can mitigate cognitive deficits, this claim relies on prior studies without elucidating the mechanisms underlying this relationship.

Furthermore, the discussion does not adequately address the limitations or propose approaches to resolve discrepancies, such as the significant variation in MDM cell counts across models. A deeper analysis of potential technical limitations and the need for future studies to standardize methodologies would lend greater credibility to these findings.

Functional validation is essential to substantiate the claim that MDM persistence contributes to cognitive deficits. This could involve targeted interventions, such as pharmacological or genetic manipulation of MDM populations, and subsequent behavioral assays to assess their impact on cognitive outcomes directly. Such experiments would provide robust evidence linking MDM persistence to cognitive impairments and strengthen the translational relevance of the findings.

Minor Comments:

- The manuscript contains numerous errors and repetitive elements that should be corrected for readability and coherence.

For instance:

To avoid redundancy, terms such as CCI, injured brain, contusion injury, and TBI should be standardized throughout the text.

Repeated definitions of these terms should be removed to streamline the content.

Addressing these issues will improve the overall quality and impact of the manuscript, ensuring clarity and methodological rigor.

Reviewer #2

(Remarks to the Author)

Very interesting and well executed study showing that infiltrated monocytes differentiate into macrophages that persist in the brain for up to 8 months. These macrophages accumulate in the pericontusional region and undergo phenotypic transition, while maintain their phagocyte competency. Using bulk RNA-seq, the authors show that these recruited macrophage exhibit long after brain engraftment a transcriptomic identity distinct from microglia that overlaps with established signatures of aging, senescence, and disease.

Comments:

Figure 1e/1f: Some controls are missing such as background controls as well as non TBI level of infiltration.

It is not clear if some MDMs become microglia or not.

Authors only analysed their populations as bulk. Single cell RNA-seq profiling will have given extra insights on the possibility of recruited cell to differentiate into microglia as well as their heterogeneity.

Reviewer #3

(Remarks to the Author)

Monocytes-derived macrophages (MDMs) play an important role in post-TBI neuroinflammation and influence neurological outcomes. This study aimed to investigate the long-term fate of peripheral monocytes after infiltrating the post-TBI brain, focusing on their potential transition to microglia and transcriptomic characteristics. A reporter mouse strain driven by CCR2-creERT2 was used, which labels proinflammatory monocytes that infiltrate the brain through chemokine-receptor interactions. Part of the findings were confirmed using Ms4a3-cre-driven reporter mice. The main finding is that MDMs persist in the brain for up to 8 months post-TBI while remaining transcriptomically distinct from microglia and exhibiting signatures associated with aging and disease.

This study addresses an important topic and has several strengths, including the use of both sexes, extended observation periods (up to 8 months), and two distinct reporter lines. While the findings are interesting, the study has significant limitations. The results are largely descriptive, lack deep functional validation, and provide insufficient spatial and heterogeneity information. For example, RNA-seq findings are not validated, and there is no functional evidence supporting the claim that MDMs causally contribute to cognitive deficits or disease. The conclusion that this work "significantly advances the understanding of long-lasting MDMs" or provides "critical knowledge for targeted therapeutic interventions for myeloid cells" feels overstated.

In specific comments:

1. In general, figures are difficult to interpret due to small text, excessive spacing, and low-resolution images. Flow cytometry plots are particularly hard to view.
2. Figure 1 measures the presence of engrafted monocytes as TdTomato+ cells in the CD11b+CD45+ population. scRNA-seq data have suggested that some brain-resident myeloid cells, such as microglia and CNS macrophages, also express CCR2. The authors reported that TdTomato+ cells were found lining in the third ventricles and choroid plexus, suggesting that they could be CNS border-associated macrophages. It is crucial to examine TdTomato+ cells in sham brains to confirm specificity. Similarly, controls showing TdTomato signals in sham and contralateral sides are needed for Figure 2.
3. Figure 2b requires higher-resolution images and high-magnification views to better visualize signal localization in each region.
4. Figure 2d shows about 30% of TdTomato+ cells were Iba1+ at 7d, which increased to about 80% at 8 months. It is surprising that such a small portion of TdT+ cells were Iba1+, given that they were CCR2+ monocytes. Nevertheless, the representative images in Figure 2c do not match this quantification; there seemed to be two TdT+ cells in the field, both Iba1+ (100%), but it is unclear which time point this represents. Full time-course data with lower magnification are necessary to validate these findings. Also, those two TdT+ cells were both positive for P2ry12 (100%), which does not represent 2e, where only less than 10% TdT+ cells are P2ry12 positive.
5. In vivo phagocytosis assay: Figure 2i claims MDM phagocytosis of beads, but the images are unconvincing. Clear images with 3D construction are needed to support this finding. Figure 2j shows that less than 10% of MDMs were associated with beads, raising questions about the meaningfulness of this functional assay.
6. The use of bulk RNA-seq limits the ability to assess spatial and heterogeneity characteristics of MDMs within the brain microenvironment. These aspects are critical for understanding their distinct roles and interactions.
7. The study provides no direct evidence that MDMs contribute to cognitive deficits or pathological outcomes. While the reported transcriptomic signature suggests associations with aging, senescence, and disease, this conclusion from high-throughput sequencing is not verified by experiments, nor does it establish causality or functional relevance.

Version 1:

Reviewer comments:

Reviewer #1

(Remarks to the Author)

In this resubmission, the authors' inclusion of an in vivo phagocytosis assay is appreciated. However, assessing phagocytic activity at 7 and 30 days post-TBI does not address the chronic inflammatory state present at 8 months, when the transcriptomic profiling was performed. Given the study's emphasis on long-term persistence of MDMs, this assay remains a readout of activity, not a test of functional relevance. It does not address necessity or sufficiency, nor does it establish a causal role for MDMs in driving chronic pathology.

As such, the revised manuscript still does not provide causal evidence linking MDMs to cognitive or pathological outcomes. The use of fate-mapped animal models presents a valuable opportunity that was not leveraged to perform mechanistic experiments, such as MDM depletion, adoptive transfer, or blockade of recruitment, which would directly test the functional role of these cells during chronic stages of TBI.

While the study contributes a long-term view of MDM transcriptomic profiles, and the use of fate mapping is technically novel within this timeframe, the work remains descriptive. The manuscript does not explain how extending fate-mapping to 8 months alters or advances current understanding of MDM biology in the context of chronic TBI. Without functional manipulation or outcome-based validation, the broader conceptual impact is limited.

Regarding the authors' response in the Methods section, the issue is not solely the choice of fate-mapping models, but

whether the Ccr2-creERT2 system introduces bias or limitations in tracking MDMs over long periods. While both Ccr2 and Ms4a3 are broadly accepted tools for macrophage fate mapping, the authors do not explain why these models are specifically suitable for long-term post-TBI MDM tracking. In particular, the rationale for using Ccr2-creERT2 over Ms4a3 is not clearly justified, and the limitations of each model, especially in the context of chronic injury and aging, are not addressed.

Furthermore, the explanation of tamoxifen-induced labeling is insufficient. Since tdTomato signal decays rapidly after labeling, falling to ~25% by one week, the use of this system for conclusions at 8 months raises concerns about underrepresentation of true MDM presence. No quantification of label retention or validation controls are provided to assess labeling stability over time.

Finally, the authors do not justify the omission of standard lineage-specific markers such as F4/80 or CD206, which are essential for distinguishing MDMs from microglia in situ. While transcriptomic data provide important insights, they do not replace the need for spatial co-localization and protein-level validation using well-established immunohistochemical markers.

Although the animal models used are well-established and the inclusion of human data is commendable, their contribution to the study's conclusions remains unclear. The role of MDMs is not shown to be functionally important, and the novelty rests primarily on technical execution rather than conceptual insight. More direct interrogation of MDM function is needed to elevate the significance of these findings.

While two-photon microscopy may present technical challenges, it has been successfully used in multiple TBI mouse models. Moreover, the authors fail to consider or discuss alternative strategies such as ex vivo imaging, longitudinal MRI, or reporter-based tracking, which could provide meaningful functional insights.

No additional validation, such as qPCR, cytokine profiling, or pathway-level analyses, was conducted to rule out the possibility that the observed transcriptional changes reflect a general injury-induced inflammatory response rather than a specific MDM-driven effect. The authors' conclusion that "since microglia don't show the same enrichment, we interpret this as MDM-specific" is logically flawed, as it relies on an absence of signal in another cell type rather than positive, cell-intrinsic evidence.

Furthermore, the study does not include qPCR, ELISA, or in situ hybridization to validate the proinflammatory gene expression patterns identified in the transcriptomic analysis, leaving this central concern unresolved.

The justification for using only the RAWM, that it is validated and widely used, does not address the need for broader cognitive assessment. The absence of complementary behavioral tests represents a missed opportunity to enhance experimental robustness and better characterize MDM contributions to functional impairment. Moreover, the lack of a defined strategy to functionally interrogate MDMs at the chronic time point further weakens the study's translational impact.

The authors refer to previous studies to justify their conclusions, but this manuscript does not build upon or experimentally test those earlier functional findings. As a result, it lacks functional insight and remains largely descriptive rather than mechanistic. While the addition of phagocytosis assessments adds some functional context, the data presented do not meaningfully inform therapeutic strategies or address the core translational gap. There is no discussion of underlying mechanisms, proposed interventions, or clearly defined future directions.

In summary, the authors attempt to justify publication by emphasizing the novelty of being the first to describe MDM persistence at 8 months post-TBI. However, without demonstrating pathological relevance, this remains a descriptive finding rather than a substantive scientific advance.

Reviewer #2

(Remarks to the Author)

The authors have addressed the issues that were raised.

Reviewer #3

(Remarks to the Author)

I thank the authors for their efforts in addressing my previous comments and for improving the manuscript. While some limitations remain—particularly that the results are largely descriptive, lack deep functional validation, and provide limited spatial or heterogeneity information—I acknowledge that this work may still represent an important initial step toward advancing our understanding of long-lasting monocyte-derived macrophages (MDMs) after traumatic brain injury (TBI).

That said, the quality of the figures continues to be a concern. Many images, especially the immunostaining panels including newly added data, are still of low resolution. Text within the figures remains small and difficult to read, and there is excessive empty space between panels that detracts from overall clarity. Further improvement in figure formatting and resolution would help ensure the data are more accessible and impactful.

REVIEWER COMMENTS

Reviewer #1 (Remarks to the Author):

This manuscript, "Fate mapping of peripherally derived macrophages reveals a long-lasting engrafted population that maintains a distinct transcriptomic profile for up to 8 months after Traumatic Brain Injury" by Paladini et al., reveals the long-term persistence and functional role of monocyte-derived macrophages (MDMs) in the brain following traumatic brain injury (TBI). Using innovative fate-mapping models, the authors show that MDMs remain distinct from microglia, retain phagocytic activity, and exhibit aging- and disease-related signatures up to 8 months post-TBI. The work addresses a critical gap in understanding MDMs' contribution to chronic neuroinflammation and cognitive deficits, providing robust evidence and potential therapeutic targets.

General Comments:

The existing literature extensively documents the neurodegenerative and detrimental roles of MDMs in the neuroinflammatory response following TBI. The macrophage and microglial cell signatures post-TBI are well-established. While characterizing this response reinforces previous findings, it offers only descriptive insights without shedding light on underlying mechanisms. Furthermore, the characterization does not convincingly demonstrate a causative link to the development of memory and learning deficits. In addition, the study's focus on MDMs in injured brains at this stage fails to provide meaningful data regarding their phagocytic or alternative functional roles. The lack of functional modulation or intervention experiments leaves it unclear whether the observed characterization results from MDM activity or reflects a chronic inflammatory environment. As such, the work does not advance understanding of the peripheral immune response to TBI or elucidate novel functional insights about MDMs. Given the extensive preclinical studies in the literature that have characterized MDMs and explored various treatments, this study does not contribute significantly novel information. The reviewer recommends incorporating studies that evaluate the role of MDMs in the context of TBI. Integrative approaches, particularly those that address current knowledge gaps and provide robust functional outcomes, would strengthen the conclusions and enhance the study's overall impact.

We appreciate the reviewer for recognizing that our work addresses a critical gap in understanding the contribution of MDMs to neuroinflammation and provides strong evidence along with potential therapeutic targets. We are also grateful for the reviewer's thoughtful feedback and valuable suggestions, which we have integrated in the resubmitted manuscript.

We agree with the reviewer that there is existing literature on MDMs in TBI. However, there remains a limited understanding of the long-term trajectories of MDMs following brain infiltration. Our study offers novel insights into the temporal dynamics and transcriptomic signatures of MDMs in the context of TBI. To our knowledge, no prior research has utilized fate mapping models to track MDM trajectories specifically in TBI, and the 8-month post-injury time point we examined remains largely unexplored in the TBI field. Additionally, we present RNA sequencing data from two complementary animal models, with analyses performed at three distinct time points post-injury, which strengthens the robustness of our findings. We also provide functional readouts, such as the phagocytosis assay, to complement the transcriptomic data. While we agree that further exploration of the underlying mechanisms of MDMs infiltration and engraftment would be valuable, this was beyond the scope of the current study.

In the revised manuscript, we have carefully addressed the reviewer's concerns as follows:

- Regarding the reviewer's concern about the causative link between MDMs and the development of memory and learning deficits, and the uncertainty of whether the observed characterization is due to MDM activity or a chronic inflammatory environment, we would like to emphasize that both our work and others have previously demonstrated a direct implication of MDMs in the development of TBI-induced memory deficits using pharmacological (Morganti et al., 2015, Morganti et al., 2016, Somebang et al., 2021) and genetic (Sample et al., 2010, Hsieh et al., 2014 Chou et al., 2018) approaches. This study expands on this foundation, specifically focusing on investigating the long-term trajectory and temporal dynamics of MDMs following TBI.

- In response to the reviewer's comment regarding the lack of meaningful data on the phagocytic or alternative functional roles of MDMs, we performed a validated *in vivo* phagocytosis assay to assess the phagocytic capacity of MDMs at 7 and 30 days post-TBI. Our data (Figure 2h-k) show that these cells retain their phagocytic ability at levels comparable to those reported in previous studies (Feng et al., 2021).

- While our current study does not include modulation or intervention experiments, we acknowledge the importance of these approaches. Our work builds upon previous studies in which MDM infiltration was blocked either genetically or pharmacologically. We believe that the findings presented here will provide a valuable foundation for future studies focused on modulating MDM activity in TBI. We have addressed this point in the conclusions and limitations section of the revised manuscript.

- In response to the suggestion for integrative approaches to address current knowledge gaps, and to further contribute to the understanding of TBI and MDMs, we have validated our findings from two different animal models by conducting single-nucleus RNA sequencing on human samples. This dataset represents the first single-nucleus RNA sequencing data from patients who suffered TBI years earlier, offering a unique and novel contribution to the field. Additionally, our MDMs data were validated using a publicly available Alzheimer's disease database (SEA-AD, <https://portal.brain-map.org/explore/seattle-alzheimers-disease>).

We hope these revisions effectively address the reviewer's concerns and enhance the manuscript's contribution to the understanding of MDMs in TBI.

Major Comments:

Abstract:

The abstract should not include general background information on the topic, as this is already well-established in the field. Instead, it should focus on the specific contributions this study makes to advancing knowledge about MDMs in the context of TBI.

The abstract has been revised based on the reviewer's suggestions, with a greater emphasis on highlighting the novel contributions of this study to the field.

Introduction:

While the role of MDMs in TBI is well-documented, it is also worth noting that nanoparticle-based drug delivery systems and vagus nerve stimulation are being investigated to modulate MDM activity, resolve neuroinflammation, and restore cognitive function post-TBI. However, this study does not explore or analyze the functional roles of MDMs following TBI, which limits its scope and impact.

We agree with the reviewer that several approaches, including nanoparticle-based drug delivery and vagus nerve stimulation, have been proposed to modulate MDM activity in TBI. These approaches are now mentioned in the revised document. The scope of the present study was not to analyze the functional role of MDMs following TBI, as this has already been explored by us and others (Morganti et al., 2015, Morganti et al., 2016, Chou et al., 2018, Somebang et al., 2021). Instead, our focus is on investigating the long-term trajectories and transcriptomic dynamics of MDMs in TBI tissues from both mice and humans.

Methods:

• The study's reliance on the Ccr2-creERT2 model with limited validation through complementary models (e.g., Ms4a3-cre) raises concerns about the generalizability of the findings. Furthermore, no clear justification for selecting these specific mouse models is provided, leaving the rationale unclear.

The Ms4a3 and Ccr2 models are two well-established leading tools for macrophage fate mapping, and have recently been utilized to track macrophages in the brain during development (Dick et al., 2022), after stroke (Chen et al., 2020), radiation (Silvin et al., 2022) and in neuroinflammation (Amorim et al., 2022). Further justification for our choice of these models is now provided in the revised introduction. While most fate mapping studies typically rely on a single model, we have employed both models in our study to validate our findings. Although the majority of our characterization is conducted using Ccr2-creER mice (as discussed in the revised limitations section), we demonstrate consistent results in both behavioral impairment (cognitive deficits measured in the RAWM) and MDMs transcriptomic signatures in Ms4a3-cre mice, which strengthens our conclusions. Importantly, in the revised version of the manuscript, we further validated the core MDMs signature using new single-nucleus RNA sequencing data from human TBI and AD patients.

• While the Ccr2-creERT2::Ai14D system achieves high initial labeling efficiency (~80–85%), the rapid turnover of Ly6Chi monocytes and the decline in labeled cells over time (~25% after one week) may lead to underrepresentation or bias in longitudinal tracking of MDMs.

We thank the reviewer for raising this important point. The advantage of using the Ccr2-creERT2::Ai14D inducible model is the ability to control the timing of labeling. By administering three out of five doses of tamoxifen prior to injury and two final doses after, we were able to specifically and efficiently label the first wave of monocyte infiltration into the brain after TBI. To our knowledge, it is not yet established whether peripheral monocytes infiltrate the TBI brain in distinct waves. In contrast, the Ms4a3-cre model is constitutive, meaning that nearly all infiltrating monocytes are labeled, regardless of the timing post-injury. We have added this key distinction in the revised manuscript.

• The study misses an opportunity to utilize cellular markers that accurately distinguish the phenotypes of microglia and macrophages, leading to potentially invalid interpretations. Key markers such as CD68 or CD206 for microglia and F4/80 for peripheral macrophages should have been used. Additionally, the immunofluorescence data should be reanalyzed to improve accuracy and clarity.

In our study, the characterization of microglia and macrophage phenotypes is based on both immunofluorescence (Iba1, P2yr12 and tdTomato) and transcriptomic analyses, which, in combination with the use of fate mapping models, allow us to unambiguously distinguish MDMs' phenotypes from microglia's.

Furthermore, the revised methods section now provides a more accurate and detailed description of the immunofluorescence data analysis process that we hope will clarify our approach and the robustness of our characterization.

- *The assertion that MDMs transition to a phenotype distinct from microglia is insufficiently supported. While markers like Iba1 and P2ry12 suggest partial differentiation, the absence of statistically significant increases in P2ry12+ expression over time undermines the claim of meaningful microglia-like adaptation. For instance, claiming that Iba1, expressed by macrophages, is also expressed by microglial cells (resident cells in most cases) lacks sufficient validation.*

We want to clarify that our study does not claim that MDMs transition into microglia. On the contrary, our findings suggest that MDMs do not completely adopt a microglia-like phenotype. This is supported by both our immunofluorescence data, which show no significant increase in P2ry12 expression over time, as well as by our RNA sequencing analysis, where MDMs maintain a distinct transcriptomic identity even 8 months post-injury. We hope this clarification addresses the reviewer's concerns, and we appreciate the opportunity to elaborate on this point.

- *The legends and results do not specify the number of animals and stats analyzed per group, the regions assessed in each animal, or the statistical methods employed. These omissions reduce the study's transparency and reproducibility.*

We thank the reviewer for pointing out these important details. In the revised version of the manuscript, we have addressed these concerns by clearly specifying the number of animals, the statistical analyses conducted for each group and the brain region assessed.

Results:

- *Extensive research has established that CCI can result in long-term cognitive deficits in mice, persisting for several months post-injury. However, the added value of a descriptive characterization at 8 months post-TBI is unclear. The mere persistence of MDMs up to 8 months post-TBI is insufficient to establish their functional relevance. Persistence alone does not imply a pathological role unless causality is definitively demonstrated. A more informative approach could involve analyzing the inflammatory response through neuroimaging at multiple time points post-TBI, extending up to 2 years.*

We thank the reviewer for their valuable feedback. We agree that we did not prove causality for the MDM and memory deficits in the current manuscript. We also acknowledge that we did not specify the choice of the 8-month time point in the original manuscript, and we have addressed this in the revised version. To the best of our knowledge, long-term (greater than 6 months) TBI-associated behavioral alterations have been characterized in only a few studies (Ritzel et al., 2020, Stelfa et al., 2022, Obenaus et al., 2023). However, no data specifically regarding MDMs has been reported chronically at 8 months after TBI, making it a novel and unexplored area. Interestingly, at 8 months post-TBI, we still observe persistent cognitive deficits in the RAWM task and a sustained population of MDMs. While we do not claim causality in this study, we hypothesize that the presence of this persistent population may contribute to the prolonged cognitive impairments, building upon

previous studies from our group and others that have established causality during the initial stages of infiltration. Moreover, although we appreciate the reviewer's suggestion of a time course neuroimaging assessment of inflammation, we believe this falls outside the scope of the current study. We have focused on the long-term presence of MDMs after TBI, and such neuroimaging analyses would require a separate and more extensive investigation.

- *Neuroimaging techniques should be employed to accurately characterize the progression of "Disease Inflammatory Macrophages" (DIM) at a chronic stage. Live imaging via two-photon microscopy to monitor macrophage infiltration and activity in the injured brain or combining imaging with systemic administration of fluorescently labeled antibodies to detect activation markers would provide more robust and functional insights. The enrichment of DIM and disease-associated microglia (DAM) signatures may indicate overlap; however, it risks confounding the interpretation of MDM-specific contributions, especially without proper functional dissection. The current characterization of microglia/macrophage populations is insufficiently precise and leads to potentially erroneous conclusions.*

We agree that neuroimaging techniques, including two-photon microscopy, could provide valuable insights into macrophage infiltration and activity in the injured brain. However, two-photon live imaging has some limitations in the context of our experiment. For example, the placement of a cranial window would present challenges in accurately capturing the full infiltration process. Additionally, while an implanted glass prism at the cavitation site could facilitate deeper imaging, it may be difficult to implement given the inflammation and scarring at the injury site. The depth of imaging with a glass window would also likely be insufficient, considering where we observed the infiltration and the number of cells involved.

Regarding the potential overlap between DIM and DAM signatures, we apologize for any confusion. Our analysis indicates that MDMs at different time points exhibit greater enrichment in both DIM and DAM signatures, as well as senescence and aging-related markers, compared to microglia. We have further clarified this distinction in the revised manuscript to avoid any misunderstandings.

- *Transcriptomic profiling does reveal distinct temporal signatures, but the overlap with aging, senescence, and disease-associated profiles may reflect a generalized response to chronic inflammation rather than effects specific to TBI.*

While it is true that the transcriptomic profiling reveals overlap of aging, senescence, and disease-associated signatures in MDMs collected at different time points after TBI, we believe that the lack of similar enrichment in microglia (which we use as a control) indicates that these profiles are not simply a generalized response. The observed specific enrichment suggests that these signatures are more closely associated with the MDMs' response to TBI, rather than a general inflammatory reaction in the tissue.

- *The statement, "Macrophages are professional phagocytes," is an intriguing observation but lacks scientific depth or relevance in the context of this study. It fails to add meaningful insight.*

In response to the reviewer's comment, we have revised the wording in the manuscript.

• The claim regarding “morphological changes associated with phagocytic marker Iba1 over time after engraftment, not fully differentiated into microglia” is problematic. Without proper characterization of the cellular phenotype, such assertions lack validity. Morphological changes, such as reduced cell volume, are more likely reflective of survival adaptations rather than functional differentiation or pathological significance.

We apologize for the confusion and thank the reviewer for their feedback. In that sentence (results, 3.3), our intent was to use the term "phagocytic" to refer to a specific cell population (phagocytes), not to make any claims about functional differentiation. The phagocytic capacity of MDMs is functionally evaluated in section 3.4 of the results (phagocytosis assay). We have rephrased that sentence in the revised manuscript to avoid any ambiguity and ensure clarity.

• Genes expressed by activated microglia or peripherally derived macrophages should be analyzed to validate RNA-seq results. Characterization of pro-inflammatory cytokines, such as IL-1 β , TNF- α , and IL-6, co-localized with Iba-1 (but not F4/80), would help distinguish pro-inflammatory microglia from macrophages.

We thank the reviewer for their suggestion. Our fate mapping mouse models already provide a clear distinction between pro-inflammatory microglia and macrophages, addressing the concern of accurately identifying these populations. Nevertheless, we agree that sequencing validation is crucial and while we did not characterize pro-inflammatory cytokines expression in microglia and MDMs, we have additionally confirmed the enrichment of the MDMs signature using a human TBI dataset, further strengthening the validity of our findings.

• Identifying 29 core genes shared by MDMs across models is intriguing but remains incomplete without functional validation. Whether these genes are causal in cognitive deficits or merely epiphenomenal to the injury response is unclear.

We appreciate the reviewer's thoughtful comment. While in our study we do not claim causality between the updated 114 core genes and cognitive deficits, we have expanded the discussion and highlighted that 31 of these core genes are part of the druggable genome (<https://dgidb.org>), a subset of genes that encode proteins capable of being targeted by pharmaceutical compounds for therapeutic purposes. This additional context aims to clarify the potential relevance of these genes in offering valuable insights for developing targeted therapeutic strategies focused on infiltrated macrophages.

• Pharmacological studies should be integrated to strengthen these findings, including the depletion of macrophages using liposome-encapsulated clodronate or CCR2 inhibitors. Then, the impact on TBI outcomes should be assessed. This approach would provide a more definitive understanding of macrophage contributions than essential gene signature characterization.

We thank the reviewer for their suggestion. As mentioned in the introduction, both our group and others have already demonstrated that pharmacological or genetic blocking of Ccr2 can prevent the development of TBI-induced cognitive deficits in mice. We have expanded on this section in the revised manuscript to provide additional context. However, it is worth noting that CCR2 antagonists have not shown success in clinical studies, highlighting the need for further understanding of the role of these cells. Such knowledge will be crucial for more precise targeting and modulation in therapeutic approaches.

- *Neutrophil populations infiltrating the brain must also be evaluated to determine their short-term, long-term, and beneficial or detrimental roles in TBI, particularly in macrophage recruitment and activity.*

We agree that neutrophils, as short-lived circulating cells that rapidly migrate to the injury site as part of the initial host defense, play an important role in the acute phase TBI, especially in recruiting monocytes (Jassam et al., 2017). However, since this is outside the scope of the current study, we did not focus on neutrophil populations. We acknowledge the reviewer's suggestion and recognize its relevance for future research in this field.

- *To elucidate the function of MDMs, the immune response pathways implicated in TBI should be validated by qPCR, ELISA, immunostaining, and other methods. Special attention should be given to genes linked to aging and senescence that are upregulated in MDMs.*

We appreciate the reviewer's suggestion regarding the validation of immune response pathways in MDMs. Although we did not use techniques such as qPCR or ELISA in this study, we strengthened our findings by employing two different fate mapping mouse models to confirm the MDM signatures. Additionally, we are incorporating a human TBI single-nucleus dataset in the revised manuscript to further validate the signatures we identified. We believe this additional validation will enhance the robustness of our conclusions.

- *Although the study highlights long-term MDM persistence and transcriptomic changes, it does not address whether modulating MDM activity or recruitment could mitigate cognitive impairments. This omission represents a critical gap in translational relevance.*

As mentioned previously, we and others have demonstrated that blocking the recruitment of MDMs can mitigate TBI-induced cognitive deficits. While we fully agree that fine modulation of MDM activity would provide important insights, we consider this a logical next step for future studies. These future experiments will build upon the foundational knowledge presented in this manuscript and will help further explore the translational relevance of MDMs modulation in TBI.

- *Behavioral Testing: The study proposes a link between cognitive deficits and MDMs but lacks direct mechanistic evidence connecting MDM-specific transcriptomic changes to cognitive impairments.*

We thank the reviewer for raising this important point. Our claims regarding the link between MDMs and cognitive deficits are built on previous work that has demonstrated a causal relationship between MDMs and cognitive impairments (as mentioned above). However, in the context of our study, we did not specifically test this causal link. We have carefully rephrased our statements in the revised manuscript to reflect this, including a sentence that explicitly addresses this point in the limitation section.

- *"Cognitive impairments in RAWM": While the radial arm water maze (RAWM) assesses spatial memory, additional cognitive domains such as executive function and attention remain untested. Exploring these domains could offer a more comprehensive understanding of TBI-induced impairments.*

• *The absence of comparisons with other TBI models or cognitive assays limits the generalizability of the findings. Incorporating other paradigms, such as the Morris Water Maze (MWM), novel object recognition, or Barnes Maze, would strengthen the study's conclusions.*

We thank the reviewer for their thoughtful suggestions. While we acknowledge the existence of other tasks that assess different aspects of cognition, we chose to use the radial arm water maze (RAWM) due to its strong validation and wide use, both by our group and others in the field, to assess learning and memory. Additionally, the primary aim of this paper is not to provide a detailed analysis of the various cognitive domains affected by TBI but rather to highlight the long-lasting deficits observed at a novel and very late/chronic time point after injury. We believe this focus aligns with the objectives of our study.

Figures

The figures should be reconstructed to enhance clarity, reduce redundancy, and effectively highlight the most significant findings. The font size and image quality should also be improved to ensure legibility. Currently, several errors and inconsistencies undermine the impact of the figures:

1. Figure 1c: The reliance on tamoxifen-induced labeling may not accurately reflect newly recruited or replenished MDM populations, especially at 30 days or 8 months post-TBI. This limitation should be addressed or acknowledged in the figure's interpretation.

We appreciate the reviewer's comment. This point has already been addressed in an earlier section of the revision, where we discuss the relevance of temporally controlled labeling of MDMs after TBI.

2. Figure 2a: The brain map shows identical representations for the anterior and central areas, which introduces ambiguity and requires correction.

We thank the reviewer for pointing this out. Upon review, we realize that this was a mistake on our part. We updated the figure accordingly.

3. Figure 2b: The representative images of the damaged cortical region do not correspond to the lesion measurements presented in Figure 2f. This discrepancy should be resolved to ensure consistency between visual and quantitative data.

We thank the reviewer for highlighting this issue. In the revised version, we have included zoomed-in images for each time point, as well as rendering examples that more accurately represent the quantifications presented in Figure 2. We believe this addresses the discrepancy and ensures consistency between the visual and quantitative data.

4. Individualized Data Points: All figures should include markers to indicate data from individual samples, enhancing transparency and allowing for a more robust interpretation of variability.

We thank the reviewer for their suggestion. In the revised manuscript, we have addressed this by representing individual samples in all figures, or in the supplementary figures when more appropriate.

5. Figure 2: The volumetric density (μm^3) measurement of tdTomato+Iba1+ cells across three selected coronal sections is overly simplistic and inappropriate for drawing conclusions about microglial/macrophage cell density. Instead, the following methods should be employed:

o *Stereological Analysis: Perform stereological assessments in the three proposed regions to accurately quantify cell density.*

o *Morphological Analysis: Sholl analysis and Neurolucida can be used to evaluate how microglial/macrophage morphology changes over time and across regions.*

We thank the reviewer for their suggestion. Imaris is a leading software for 3D image analysis and visualization and volumetric reconstruction (Shebestari et al., 2022, Feichtenbiner et al, 2025) and its machine learning-based segmentation improves segmentation accuracy and provides quantitative insights into cellular structures and processes for reproducible stereological assessments. We believe this approach offers a more comprehensive analysis compared to alternative software such as Neurolucida, which is designed for neuronal tracing. Additionally, images were captured at an appropriate magnification for the current analysis, and we believe a Sholl analysis would not be suitable for this dataset. To clarify, we have updated the methods section to include a more detailed description of the analysis approach and parameters used.

Discussion

To contextualize this study's findings, it is necessary to thoroughly discuss the previous literature on central and peripheral inflammation following TBI in both preclinical and clinical studies. This would help clarify what the study uniquely contributes to understanding the causes of behavioral deficits, particularly cognitive impairments. While the study asserts that preventing MDM infiltration can mitigate cognitive deficits, this claim relies on prior studies without elucidating the mechanisms underlying this relationship.

Furthermore, the discussion does not adequately address the limitations or propose approaches to resolve discrepancies, such as the significant variation in MDM cell counts across models. A deeper analysis of potential technical limitations and the need for future studies to standardize methodologies would lend greater credibility to these findings.

Functional validation is essential to substantiate the claim that MDM persistence contributes to cognitive deficits. This could involve targeted interventions, such as pharmacological or genetic manipulation of MDM populations, and subsequent behavioral assays to assess their impact on cognitive outcomes directly. Such experiments would provide robust evidence linking MDM persistence to cognitive impairments and strengthen the translational relevance of the findings.

We thank the reviewer for these insightful comments. In the revised manuscript, we have expanded the discussion to include a more thorough review of the previous literature on MDMs in TBI, to highlight the novelty of our work. Additionally, we have incorporated a section discussing the limitations of our study, including the variability across models, and highlighted the need for future research to address the technical and knowledge gaps. We also appreciate the reviewer's suggestion regarding functional validation. While we did not perform such experiments in this study, we have noted this as an important area for future investigation to further substantiate the relationship between MDM persistence and cognitive deficits. We believe that these future investigations will greatly benefit from the discoveries presented in the current paper.

Minor Comments:

• *The manuscript contains numerous errors and repetitive elements that should be corrected for readability and coherence. For instance:*

To avoid redundancy, terms such as CCI, injured brain, contusion injury, and TBI should be standardized throughout the text.

Repeated definitions of these terms should be removed to streamline the content.

Addressing these issues will improve the overall quality and impact of the manuscript, ensuring clarity and methodological rigor.

We appreciate the reviewer's suggestion regarding the redundancy of terms and have made efforts to standardize the terminology throughout the revised manuscript, using "TBI" consistently, except when describing the injury in the methods section and the pericontusional area in the results.

Reviewer #2 (Remarks to the Author):

Very interesting and well executed study showing that infiltrated monocytes differentiate into macrophages that persist in the brain for up to 8 months. These macrophages accumulate in the pericontusional region and undergo phenotypic transition, while maintain their phagocyte competency. Using bulk RNA-seq, the authors show that these recruited macrophage exhibit long after brain engraftment a transcriptomic identity distinct from microglia that overlaps with established signatures of aging, senescence, and disease.

We are glad that the reviewer found the study interesting and well-executed, and we truly appreciate their thoughtful comments that we addressed below.

Comments:

Figure 1e/1f: Some controls are missing such as background controls as well as non TBI level of infiltration.

We thank the reviewer for this comment. MDMs infiltration data in non TBI mice (sham) is shown by immunofluorescence in the supplementary materials (Supplementary Figure 4a). Additionally, we have included flow cytometry data to further support our findings showing the absence of MDM infiltration in control mice (Supplementary Figure 1d).

It is not clear if some MDMs become microglia or not.

We thank the reviewer for highlighting this important point. To clarify, our study does not suggest that MDMs transform into microglia. Rather, our data indicate that MDMs do not fully adopt a microglia-like phenotype. This is evident from our immunofluorescence data, which show no significant increase in P2ry12 expression over time, as well as from our RNA sequencing analysis, where MDMs maintain a distinct transcriptomic profile even 8 months post-injury. We hope this clears up any confusion, and we have revised the manuscript to provide a more detailed explanation of this aspect.

Authors only analysed their populations as bulk. Single cell RNA-seq profiling will have given extra insights on the possibility of recruited cell to differentiate into microglia as well as their heterogeneity.

We appreciate the reviewer's input. We agree that single-cell RNA-seq profiling would provide additional insights. To address this, we have validated our core MDM signature in a newly created human TBI single nuc RNA-seq dataset, which we believe can offer further insights into this aspect.

Reviewer #3 (Remarks to the Author):

Monocytes-derived macrophages (MDMs) play an important role in post-TBI neuroinflammation and influence neurological outcomes. This study aimed to investigate the long-term fate of peripheral monocytes after infiltrating the post-TBI brain, focusing on their potential transition to microglia and transcriptomic characteristics. A reporter mouse strain driven by CCR2-creERT2 was used, which labels proinflammatory monocytes that infiltrate the brain through chemokine-receptor interactions. Part of the findings were confirmed using Ms4a3-cre-driven reporter mice. The main finding is that MDMs persist in the brain for up to 8 months post-TBI while remaining transcriptomically distinct from microglia and exhibiting signatures associated with aging and disease.

This study addresses an important topic and has several strengths, including the use of both sexes, extended observation periods (up to 8 months), and two distinct reporter lines. While the findings are interesting, the study has significant limitations. The results are largely descriptive, lack deep functional validation, and provide insufficient spatial and heterogeneity information. For example, RNA-seq findings are not validated, and there is no functional evidence supporting the claim that MDMs causally contribute to cognitive deficits or disease. The conclusion that this work "significantly advances the understanding of long-lasting MDMs" or provides "critical knowledge for targeted therapeutic interventions for myeloid cells" feels overstated.

We genuinely appreciate the reviewer's recognition of the strengths of our work. Though our study may appear descriptive, it offers novel insights into the dynamics and transcriptomic profiles of MDMs in the context of TBI. Fate mapping models have not been previously used to track MDMs trajectories in TBI, and the 8-month post-injury time point we investigated remains a largely underexplored area in TBI research. Additionally, our RNA sequencing data, derived from two complementary animal models and analyzed at several time points after injury, provides added depth and reliability to our findings. Beyond the characterization, we also include functional data, such as the phagocytosis assay, to support our conclusions. While our current study does not include an experiment explicitly designed to demonstrate the causal contribution of MDMs to cognitive deficits or disease, we base our work on a substantial body of prior research, including studies from our lab and others, that have consistently shown a causal relationship between MDMs and cognitive deficits in TBI, described in the introduction (Hsies et al., 2014, Morganti et al., 2015, Morganti et al., 2016, Chou et al., 2018, Somebang et al., 2021). Additionally, we have now further validated our core MDM signature by integrating a newly generated transcriptomic dataset from human TBI samples in the revised manuscript. In response to the reviewer's concerns about overstatements, we truly believe our work makes an important contribution to advancing the understanding of the long-term trajectories of MDMs. Moreover, the signature we identify, now validated in a new human dataset, could serve as a useful tool for future research aimed at modulating or targeting MDMs in disease and injury, helping to guide the development of therapeutic strategies. On this line, we highlighted that 31 of these core genes are part of the druggable genome (<https://dgidb.org>), a subset of genes that encode proteins capable of being targeted by pharmaceutical compounds for therapeutic purposes. Additionally, we have incorporated a section discussing the limitations of our study. We hope the reviewer will consider our revised manuscript to be appropriate and thorough.

In specific comments:

1. In general, figures are difficult to interpret due to small text, excessive spacing, and low-resolution images. Flow cytometry plots are particularly hard to view.

We thank the reviewer for their feedback. We have revised the figures to improve clarity, with larger text, reduced spacing, and higher-resolution images. We hope the updated figures are now easier to interpret.

2. Figure 1 measures the presence of engrafted monocytes as TdTomato+ cells in the CD11b+CD45+ population. scRNA-seq data have suggested that some brain-resident myeloid cells, such as microglia and CNS macrophages, also express CCR2. The authors reported that TdTomato+ cells were found lining in the third ventricles and choroid plexus, suggesting that they could be CNS border-associated macrophages. It is crucial to examine TdTomato+ cells in sham brains to confirm specificity. Similarly, controls showing TdTomato signals in sham and contralateral sides are needed for Figure 2.

We thank the reviewer for raising this important point. While we acknowledge that published sequencing data suggest that CCR2 can be expressed by microglia and CNS-associated macrophages, we have specifically addressed this concern by demonstrating the absence of TdTomato+ cells in sham CCR2-creERT::Ai14dT mice, as shown in the Immunofluorescence and flow cytometry data provided in the supplementary materials (Supplementary Figure 1 and 4). This finding supports the conclusion that the TdTomato+ cells observed in TBI mice are indeed MDMs, rather than microglia or CNS/border-associated macrophages. Furthermore, in response to the reviewer's suggestion, we have now included additional immunofluorescence images of the contralateral side in the supplementary material (Supplementary Figure 4b), which confirm that MDMs do not migrate to the contralateral hemisphere.

3. Figure 2b requires higher-resolution images and high-magnification views to better visualize signal localization in each region.

According to the reviewer's suggestion, we have now included zoomed-in regions of interest of MDMs localized in the cavitation area, choroid plexus, and third ventricle at 7, 30 days and 8 months after TBI. We hope that these additions, along with higher resolution images, will improve the overall visualization and appreciation of the results.

4. Figure 2d shows about 30% of TdTomato+ cells were Iba1+ at 7d, which increased to about 80% at 8 months. It is surprising that such a small portion of TdT+ cells were Iba1+, given that they were CCR2+ monocytes. Nevertheless, the representative images in Figure 2c do not match this quantification; there seemed to be two TdT+ cells in the field, both Iba1+ (100%), but it is unclear which time point this represents. Full time-course data with lower magnification are necessary to validate these findings. Also, those two TdT+ cells were both positive for P2ry12 (100%), which does not represent 2e, where only less than 10% TdT+ cells are P2ry12 positive.

We agree with the reviewer's comment and appreciate the constructive feedback. In response, we have modified Figure 2 by adding additional images and renderings for each time point that are more representative

of the data, particularly in terms of marker colocalization and cell volume. We believe these updated images better align with the quantification and provide a clearer depiction of the findings.

5. In vivo phagocytosis assay: Figure 2i claims MDM phagocytosis of beads, but the images are unconvincing. Clear images with 3D construction are needed to support this finding. Figure 2j shows that less than 10% of MDMs were associated with beads, raising questions about the meaningfulness of this functional assay.

We thank the reviewer for the suggestion. In accordance, we have now added representative 3D reconstruction images and an analysis workflow to Figure 2h-k to better support our findings on MDM phagocytosis. Additionally, we have expanded the description of this assay in the results section to provide more clarity. Regarding the percentage of MDMs associated with beads, we would like to highlight that this result is consistent with our previous observations in brain-engrafted macrophages that populate the brain after whole-brain radiotherapy and microglia depletion (Feng et al., 2021). We hope these revisions help address your concerns and provide stronger support for the functional relevance of this assay.

6. The use of bulk RNA-seq limits the ability to assess spatial and heterogeneity characteristics of MDMs within the brain microenvironment. These aspects are critical for understanding their distinct roles and interactions.

We agree with the reviewer that bulk RNA-seq, although providing a comprehensive overview of MDMs transcriptome, has some critical limitations. To address this, we validated our bulk RNA-seq data from mice with a newly generated single-nuc RNA-seq dataset from patients who have experienced TBI, to show enrichment of the core MDMs signature in human tissue. While we acknowledge that this approach does not provide spatial resolution, we believe it significantly strengthens our findings and robustness of our study.

7. The study provides no direct evidence that MDMs contribute to cognitive deficits or pathological outcomes. While the reported transcriptomic signature suggests associations with aging, senescence, and disease, this conclusion from high-throughput sequencing is not verified by experiments, nor does it establish causality or functional relevance.

We appreciate the reviewer's comment. As mentioned in the introduction, direct evidence of the involvement of MDMs in TBI-induced cognitive deficits comes from previous studies from our lab and others, which demonstrate that blocking MDM infiltration (either pharmacologically or genetically) prevents the development of TBI-induced cognitive deficits in mice (Sample et al., 2010, Hsieh et al., 2014, Morganti et al., 2015, Morganti et al., 2016, Chou et al., 2018, Somebang et al., 2021). Our study builds on this established knowledge and provides new insights into the long-term trajectories and dynamics of MDMs, along with their transcriptomic dynamics. The aging and disease-related transcriptomic signatures we identify are validated in two complementary fate mapping models. While we do not directly establish causality between MDMs' transcriptomic changes and long-lasting cognitive deficits in TBI mice, we speculate that these signatures may help explain the role of MDMs in cognitive impairments. We believe the data we present (including the functional assessment of MDMs phagocytic capability) offers valuable and relevant information to better understand the role of these cells in the context of TBI.

REVIEWER COMMENTS

Reviewer #1 (Remarks to the Author):

In this resubmission, the authors' inclusion of an in vivo phagocytosis assay is appreciated. However, assessing phagocytic activity at 7 and 30 days post-TBI does not address the chronic inflammatory state present at 8 months, when the transcriptomic profiling was performed. Given the study's emphasis on long-term persistence of MDMs, this assay remains a readout of activity, not a test of functional relevance. It does not address necessity or sufficiency, nor does it establish a causal role for MDMs in driving chronic pathology. As such, the revised manuscript still does not provide causal evidence linking MDMs to cognitive or pathological outcomes.

We appreciate the reviewer's feedback and valuable insights. We agree that the in vivo phagocytosis assay at 7 and 30 days post-TBI serves as a measure of subacute/chronic MDM activity and does not directly reflect their functionality at 8 months or establish per se a causal role in driving TBI-induced long-term cognitive deficits and pathology. The primary objective of this study is to demonstrate the long-term persistence of MDMs in the TBI mouse brain, as their involvement in the development of TBI-induced memory deficits has been previously demonstrated by us and others (Morganti et al., 2015, Morganti et al., 2016, Somebang et al., 2021, Sample et al., 2010, Hsieh et al., 2014, Chou et al., 2018). We have further revised the manuscript to clearly articulate these distinctions and limitations, ensuring our claims are precisely aligned with the scope of the presented data (see below).

The use of fate-mapped animal models presents a valuable opportunity that was not leveraged to perform mechanistic experiments, such as MDM depletion, adoptive transfer, or blockade of recruitment, which would directly test the functional role of these cells during chronic stages of TBI. While the study contributes a long-term view of MDM transcriptomic profiles, and the use of fate mapping is technically novel within this timeframe, the work remains descriptive. The manuscript does not explain how extending fate-mapping to 8 months alters or advances current understanding of MDM biology in the context of chronic TBI. Without functional manipulation or outcome-based validation, the broader conceptual impact is limited.

We appreciate the reviewer's insightful comments and agree that, in the current study, we did not fully leverage the potential of the fate-mapping model to functionally interrogate the role of MDMs during the chronic stages of TBI. The main novelty of our work lies in demonstrating the long-term persistence of MDMs in the injured brain up to 8 months after injury, and in providing a longitudinal transcriptomic characterization of these cells in the context of TBI. In response to the reviewer's suggestion, we have acknowledged these limitations and outlined future directions where the model could be used for functional manipulation to more directly assess the roles of MDMs in chronic TBI (see below).

Regarding the authors' response in the Methods section, the issue is not solely the choice of fate-mapping models, but whether the Ccr2-creERT2 system introduces bias or limitations in tracking MDMs over long periods. While both Ccr2 and Ms4a3 are broadly accepted tools for macrophage

fate mapping, the authors do not explain why these models are specifically suitable for long-term post-TBI MDM tracking. In particular, the rationale for using Ccr2-creERT2 over Ms4a3 is not clearly justified, and the limitations of each model, especially in the context of chronic injury and aging, are not addressed.

We agree with the Reviewer that both the Ccr2-creERT2 and Ms4a3-cre models have different modalities and intrinsic biases in tracking MDMs over long periods. We acknowledge that the Ccr2-creERT2 system relies on tamoxifen activation, which leads to time-restricted labeling of MDMs, whereas the Ms4a3 constitutive Cre model labels all MDMs, including both the first wave of infiltration and those that engraft later. Both of these models are state-of-the-art tools for linear tracing in conditions where peripheral infiltration is followed by tissue engraftment, which is a hallmark of the TBI response. We believe that the use of either model is valid in this context. Additionally, we do not believe we specifically preferred the Ccr2-creERT2 model over the Ms4a3-cre model, as most of the results observed in the Ccr2 system have been corroborated by findings in the Ms4a3-cre mice. In response to the Reviewer's comment, we have added a more detailed description of the limitations of each model in the "Conclusions, study limitations and future directions" section of the manuscript, which we hope will provide clarity on these considerations (see below).

Furthermore, the explanation of tamoxifen-induced labeling is insufficient. Since tdTomato signal decays rapidly after labeling, falling to ~25% by one week, the use of this system for conclusions at 8 months raises concerns about underrepresentation of true MDM presence. No quantification of label retention or validation controls are provided to assess labeling stability over time.

We thank the Reviewer for the thoughtful feedback. We recognize that tamoxifen-induced labeling could potentially lead to an underrepresentation of true MDM presence, particularly if subsequent waves of infiltration occur after the first week post injury. We believe we have partially addressed this limitation by incorporating the complementary Ms4a3-cre model. This model allows for continuous labeling and tracing of MDMs, thus overcoming the time-restricted nature of the tamoxifen-induced system and providing a broader view of MDM dynamics over time. Furthermore, we validated the accuracy of our labeling system by showing that control (sham) mice do not exhibit any engrafted tdTomato+ MDMs, which supports the reliability of our approach. While we did not directly assess tdTomato labeling stability over extended time points, we relied on Ai14dT mice, a well-established and robust reporter line that is widely regarded as one of the most stable and reliable systems for genetic labeling (Madisen et al., 2010). Nevertheless, we have further discussed the limitations of our study in the revised manuscript (see below).

Finally, the authors do not justify the omission of standard lineage-specific markers such as F4/80 or CD206, which are essential for distinguishing MDMs from microglia in situ. While transcriptomic data provide important insights, they do not replace the need for spatial co-localization and protein-level validation using well-established immunohistochemical markers.

We thank the reviewer for raising this point. In our study, we primarily relied on fate mapping to distinguish MDMs from microglia, as it allows for precise identification of MDMs based on their lineage tracing. We agree that additional validation using standard lineage-specific markers would strengthen

the data, particularly for spatial analysis. We have now added this point to the limitations section of the manuscript (see below).

Although the animal models used are well-established and the inclusion of human data is commendable, their contribution to the study's conclusions remains unclear. The role of MDMs is not shown to be functionally important, and the novelty rests primarily on technical execution rather than conceptual insight. More direct interrogation of MDM function is needed to elevate the significance of these findings.

We agree with the Reviewer and acknowledge that our study has a descriptive focus, but we believe it provides crucial foundational data that will inform future research focused on manipulating MDMs to explore their functional importance. By establishing a reliable system for tracing and characterizing MDMs, we hope to enable more targeted studies that can directly address their role in disease processes. We have stated this limitation in the revised manuscript.

While two-photon microscopy may present technical challenges, it has been successfully used in multiple TBI mouse models. Moreover, the authors fail to consider or discuss alternative strategies such as ex vivo imaging, longitudinal MRI, or reporter-based tracking, which could provide meaningful functional insights. No additional validation, such as qPCR, cytokine profiling, or pathway-level analyses, was conducted to rule out the possibility that the observed transcriptional changes reflect a general injury-induced inflammatory response rather than a specific MDM-driven effect. The authors' conclusion that "since microglia don't show the same enrichment, we interpret this as MDM-specific" is logically flawed, as it relies on an absence of signal in another cell type rather than positive, cell-intrinsic evidence. Furthermore, the study does not include qPCR, ELISA, or in situ hybridization to validate the proinflammatory gene expression patterns identified in the transcriptomic analysis, leaving this central concern unresolved.

While we believe the RNA-seq data presented in this work from sorted MDMs and microglia is robust and reliable, we agree with the Reviewer and value the importance of validation using qPCR, ELISA and in situ hybridization. We have addressed this point in the revised manuscript.

The justification for using only the RAWM, that it is validated and widely used, does not address the need for broader cognitive assessment. The absence of complementary behavioral tests represents a missed opportunity to enhance experimental robustness and better characterize MDM contributions to functional impairment.

We agree that the RAWM captures only a subset of the complex cognitive alterations induced by TBI. However, cognitive deficits induced by the controlled cortical impact TBI model have been previously characterized by us and others (Krukowski et al., 2018, Tucker et al., 2016, Chou et al., 2017). In line with the reviewer's feedback, we have now acknowledged the value of broader behavioral assessments in the revised document (see below).

Moreover, the lack of a defined strategy to functionally interrogate MDMs at the chronic time point further weakens the study's translational impact. The authors refer to previous studies to justify their conclusions, but this manuscript does not build upon or experimentally test those earlier functional findings. As a result, it lacks functional insight and remains largely descriptive rather than mechanistic. While the addition of phagocytosis assessments adds some functional context, the data presented do not meaningfully inform therapeutic strategies or address the core translational gap. There is no discussion of underlying mechanisms, proposed interventions, or clearly defined future directions. In summary, the authors attempt to justify publication by emphasizing the novelty of being the first to describe MDM persistence at 8 months post-TBI. However, without demonstrating pathological relevance, this remains a descriptive finding rather than a substantive scientific advance.

We thank the Reviewer for their thoughtful and constructive feedback. We agree that our study lacks functional manipulations and that it is not focused on proposing specific interventions. Rather, our intention is to provide foundational data that can inform future research aimed at the precise modulation of MDM activity. This approach will hopefully help guide the development of more targeted therapeutic strategies.

We have expanded the "Conclusions, study limitations and future directions" section of the manuscript to incorporate the Reviewer's feedback:

- *"First, the two fate-mapping models used in this study have inherent constraints. The Ccr2-CreERT2 system requires tamoxifen-induced activation, resulting in time-restricted labeling that may underrepresent the full population of MDMs infiltrating after TBI. In contrast, the Ms4a3-Cre constitutive model labels all MDMs, including both early and late infiltrating populations, but lacks temporal specificity, which limits insight into the dynamics of infiltration and engraftment."*
- *"Additionally, while previous studies from our lab and others have demonstrated a direct causal relationship between MDMs and TBI-induced cognitive deficits^{14,16,18,19,20,21,22,23}, we did not replicate these specific experiments in the current study. Instead, our work builds on the foundation of these earlier findings and takes a primarily descriptive approach, focusing on tracing the long-term trajectories and temporal transcriptomic dynamics of MDMs in TBI, a field that's unexplored and important for dementia."*
- *"Differences in labeling strategies between the two fate-mapping systems, partial overlap in analyses, and the lack of direct assessment of tdTomato labeling stability over extended time points should all be considered when interpreting the results."*
- *"Moreover, the in vivo phagocytosis assay was performed only at 7 and 30 days post-TBI, providing useful information of subacute and early chronic MDM activity, but not direct insight into the functional state or contribution of MDMs at the 8-month post-infiltration time point."*
- *"This study lacks additional validation through qPCR, in situ hybridization, or staining with standard lineage-specific markers such as CD206 and F4/80. Including these approaches in future work could strengthen the findings and provide valuable spatial context for MDM localization."*

- *“Although TBI-induced cognitive deficits have been characterized using other behavioral tasks by our group and others (Krukowski et al., 2018, Tucker et al., 2016, Chou et al., 2017), the present study focuses exclusively on evaluating learning and memory using the RAWM task.”*

We trust that the revisions address the Reviewer’s concerns, and we hope the updated manuscript meets your expectations.

Reviewer #2 (Remarks to the Author):

The authors have addressed the issues that were raised.

We thank the Reviewer for their time and thoughtful feedback. We appreciate their acknowledgment that the issues have been addressed, and we are grateful for their contribution to improving the quality of our manuscript.

Reviewer #3 (Remarks to the Author):

I thank the authors for their efforts in addressing my previous comments and for improving the manuscript. While some limitations remain—particularly that the results are largely descriptive, lack deep functional validation, and provide limited spatial or heterogeneity information—I acknowledge that this work may still represent an important initial step toward advancing our understanding of long-lasting monocyte-derived macrophages (MDMs) after traumatic brain injury (TBI). That said, the quality of the figures continues to be a concern. Many images, especially the immunostaining panels including newly added data, are still of low resolution. Text within the figures remains small and difficult to read, and there is excessive empty space between panels that detracts from overall clarity. Further improvement in figure formatting and resolution would help ensure the data are more accessible and impactful.

We sincerely thank the reviewer for their thoughtful comments and for recognizing the potential value of our study despite its limitations. We appreciate your acknowledgment of our efforts to improve the manuscript. Regarding the figures, we believe the Reviewer found the figures to be low resolution because they were compressed into PNG format and embedded within the text to manage file size. We have now uploaded the higher resolution figure files separately. Upon acceptance, we are confident we can provide uncompressed images to ensure proper visualization of the data, including improved legibility of figure text.